# Reconstitution of circadian clock in synthetic cells reveals principles of timekeeping

Alexander Zhan Tu Li[1], Andy LiWang [2] & Anand Bala Subramaniam [1] ✉

The cyanobacterial circadian clock maintains remarkable precision and synchrony, even in cells with femtoliter volumes. Here, we reconstitute the KaiABC post-translational oscillator (PTO) in giant unilamellar vesicles (GUVs) to investigate underlying mechanisms of this fidelity. We show that our encapsulation methodology replicates native protein variability. With long-term, single-vesicle tracking of circadian rhythms using fluorescent KaiB and confocal microscopy, we find that oscillator fidelity decreases with lower protein levels and smaller vesicle sizes. KaiB membrane association, observed in cyanobacteria, was recapitulated in GUV membranes. A mathematical model incorporating protein stoichiometry limitations suggests that high expression of PTO components and associated regulators (CikA and SasA) buffers stochastic variations in protein levels. Additionally, while the transcription-translation feedback loop contributes minimally to overall fidelity, it is essential for maintaining phase synchrony. These findings demonstrate synthetic cells capable of autonomous circadian rhythms and highlight a generalizable strategy for dissecting emergent biological behavior using minimal systems.

To the best of our knowledge, there are no reports of the wild-type unicellular cyanobacterium, *Synechococcus elongatus*, failing to display circadian rhythms under physiological conditions. Indeed, the circadian clock of *S. elongatus* is highly robust with near-perfect fidelity and phase synchrony despite their tiny 2 fL cell volumes[1] and lack of intercellular coupling[2–4]. The cyanobacterial circadian clock is comprised of a transcription-translation feedback loop (TTFL) and a post-translational oscillator (PTO). The PTO is composed of three core proteins—KaiA, KaiB, and KaiC—and accessory proteins SasA and CikA. The three Kai proteins, PTO proteins, that compose the core PTO generate a circadian oscillation of KaiC phosphorylation autonomously for several days when reconstituted in vitro under bulk solution conditions[5–7]. The CikA and SasA proteins enhance the robustness of the PTO by compensating for levels of KaiA and KaiB that would be too low for the PTO to oscillate on its own[8].

Much has been learned about the cyanobacterial circadian clock by reconstituting it in vitro [5–10]. However, in vitro studies are carried out in volumes of around 100 μL, which are ~10–11 orders of magnitude larger than that of a 2 fL *S. elongatus* cell. Thus, important questions regarding the cyanobacterial clock cannot be approached using current in vitro methodology. For example, it remains unclear how the *S. elongatus* clock exhibits exceptional fidelity in vivo despite their tiny interiors. The relative importance and role of the PTO and the TTFL in timekeeping is also unknown. Approaching this question in vivo is difficult because of the coupling between the PTO and the TTFL. Although attempts were made to uncouple the PTO and TTFL in vivo, the observable was the clock-controlled expression of a yellow fluorescent protein (YFP)[4,11], which is an indirect indicator of the PTO oscillation.

To obtain direct mechanistic insights into the function of the cyanobacterial circadian clock in cell-like volumes, while avoiding complicating issues associated with in vivo studies, here we encapsulated the reconstituted PTO inside artificial giant unilamellar vesicles (GUVs) to create PTO-GUVs (Fig. 1). The proteins in the GUVs partitioned with cell-like variability[12,13]. We use the PTO-GUVs to directly observe the oscillation of the PTO over a range of KaiABC

---

[1]Department of Bioengineering, University of California, Merced, CA, USA. [2]Department of Chemistry and Biochemistry, University of California, Merced, CA, USA. ✉e-mail: asubramaniam@ucmerced.edu

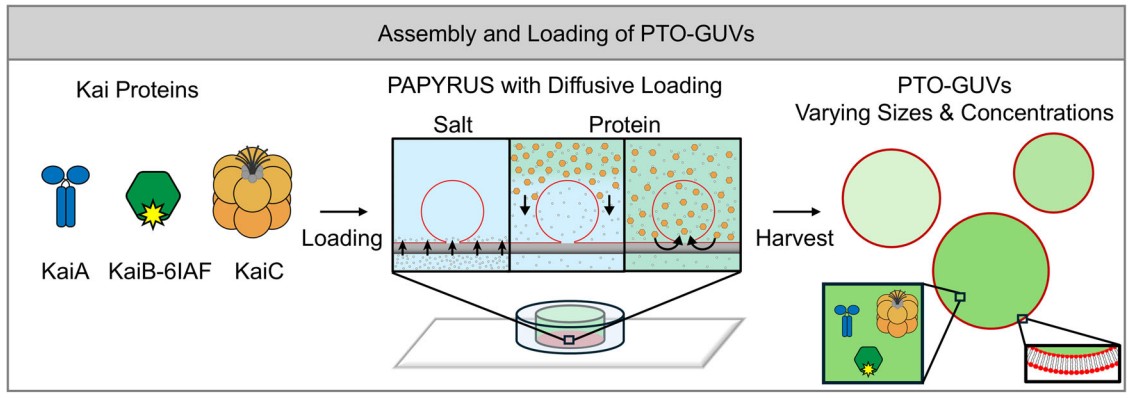

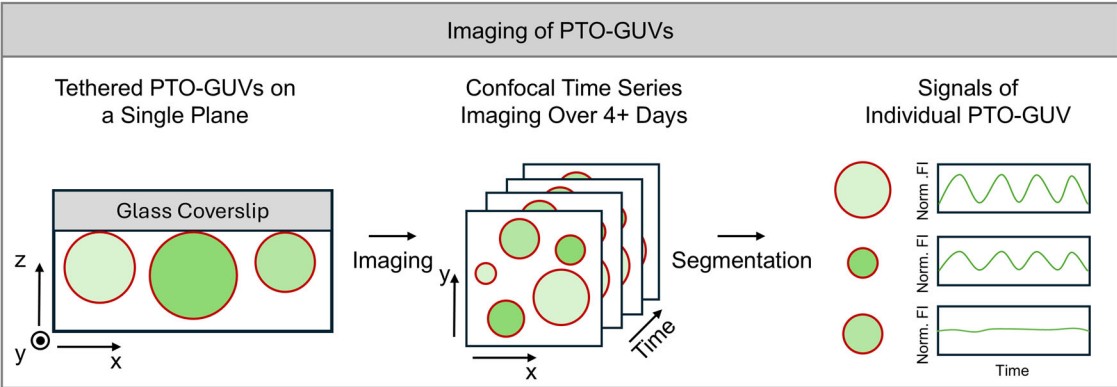

**Fig. 1 | Schematics of the experimental workflow for preparation, imaging, and analysis of PTO-GUVs.** Kai proteins that comprise the post-translational oscillator (PTO) are encapsulated into GUVs using the PAPYRUS with diffusive loading method (PAPYRUS-wDL) forming PTO-GUVs with varying sizes and PTO concentrations (red circles with green shading). The PTO-GUVs are tethered to a glass coverslip through streptavidin-biotin interactions for timelapse imaging over 4+ days with a confocal microscope. The images are then segmented to identify individual PTO-GUVs, and the time traces of the normalized fluorescence intensity (Norm. FI) is obtained.

concentrations and vesicle sizes. Using fluorescently labeled KaiB[14] and confocal microscopy, we were able to measure the circadian rhythms generated by thousands of PTO-GUVs at the single-GUV level for several days. We find that the PTO-GUVs show binary behavior, oscillating or non-oscillating, with a 4-hour range of the maximum and minimum periods in the oscillating PTO-GUVs. To capture this behavior, we define a "clock fidelity" parameter. The fidelity is the number of oscillating PTO-GUVs divided by the total number of PTO-GUVs. A fidelity of zero would mean no PTO-GUVs oscillated and a fidelity of one would mean all of them oscillated. While the fidelity decreased with decreasing GUV size and PTO concentrations, the distribution of periods in the oscillating PTO-GUVs was independent of GUV size and mean PTO concentration. We find that the highest fraction of oscillating PTO-GUVs corresponded to the approximate concentration of the PTO proteins measured in vivo. Interestingly, the GUV system recapitulates KaiB binding to membranes reminiscent of how KaiB binds to cyanobacterial membranes in vivo.

A phenomenological mathematical model that incorporates 1) cell-like partitioning of PTO proteins into GUVs, 2) concentration-based rules for the phase and amplitude of the oscillations, and 3) KaiB-membrane interactions, reproduces the measured experimental fidelity of PTO-GUVs as a function of PTO concentration and GUV size. We show that extending this model to include the buffering effect of CikA and SasA for low levels of KaiA and KaiB proteins, respectively, can explain the fidelity of the clock seen in vivo. Thus, our model shows that PTO protein concentrations in cells are tuned to maintain oscillations in individual bacteria across a clonal population with SasA and CikA support. The model also reproduced the expected variation in period and amplitude in single bacteria due to variations in protein concentrations. Although PTOs in individual simulated bacteria continue to show circadian timing, the 4-hour variation in periods due to the variation in protein concentrations leads to a loss of population synchrony over the course of four days. The loss of synchrony is reminiscent of observations in vivo where the TTFL loop was abrogated[11]. Incorporating a simplified TTFL in our model recapitulated the high degree of synchrony seen in WT cyanobacteria. Thus, high expression levels of the PTO proteins along with expression of SasA and CikA are necessary for maintaining PTO oscillations across the population while the TTFL is necessary to maintain phase synchrony across the population.

## Results

### Diffusive loading of proteins into GUVs leads to cell-like variation in intervesicular protein concentrations

Various concentrations of bovine serum albumin-fluorescein isothiocyanate conjugate (FITC-BSA) were encapsulated in GUVs prepared through the PAPYRUS-wDL (paper-abetted amphiphile hydration in aqueous solution[15] with diffusive loading) technique coupled with diffusive loading of salts and proteins[16] (see "Methods" for details). The technique produces a high yield of GUVs encapsulating salty buffers with polydisperse sizes. By selecting sub-populations of GUVs with desired characteristics, we could simultaneously study the effect of spatial parameters such as compartment size and the effect of protein concentrations. Using confocal microscopy, we found that the variation of protein concentrations obtained from diffusive loading within a population of GUVs had a coefficient of variation (CV) of ~0.3 and could be described by a gamma distribution (Fig. 2A) within a concentration range of 0.88 to 4.5 μM. This value was similar to that found in cyanobacteria and other cells in that the intercellular protein distributions could be described by a gamma distribution with CVs

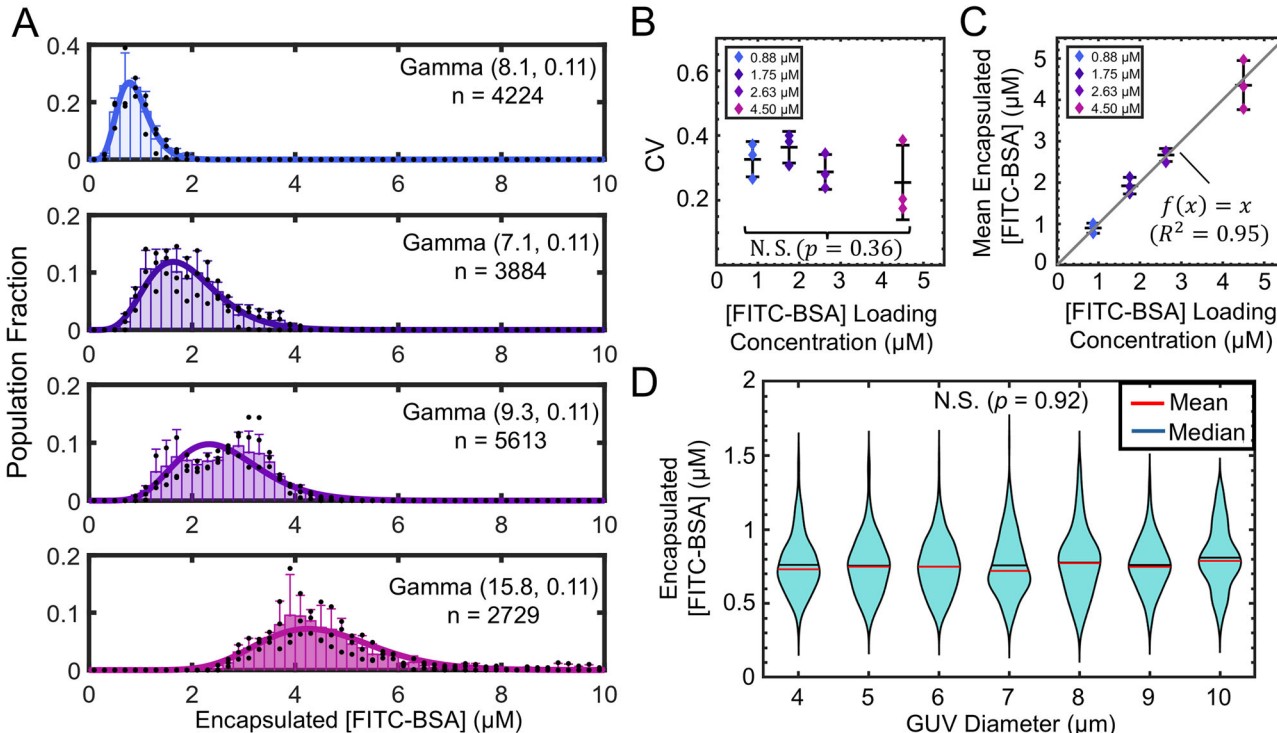

**Fig. 2 | PAPYRUS-wDL produces GUVs encapsulating functional proteins with cell-like distributions of protein concentration. A** Histograms of protein concentration distribution in GUVs prepared with loading concentrations of 0.88 μM, 1.75 μM, 2.63 μM, and 4.50 μM FITC-BSA. The bin widths are 0.2 μM. Each bar is an average of $N = 3$ independent repeats (black dots), and the error bars are one standard deviation from the mean. The distribution of protein concentrations can be described well with a gamma distribution (thick solid lines). The inset numbers, Gamma $(\alpha, \theta)$ is the best-fit gamma distribution with shape parameter, $\alpha$, and scale parameter, $\theta$, and $n$ is the total number of GUVs from the three independent samples for each concentration. **B** A one-sided ANOVA test shows that the mean coefficient of variation (CV) of the encapsulated protein did not vary with loading concentration ($p = 0.36$, not significant (N.S.)). **C** The mean concentrations of

protein in the lumen of the GUVs is linearly correlated with the loading concentrations. The black line is a linear regression ($f(x) = x$, $R^2 = 0.95$). For (**B**) and (**C**), the middle black bar shows the mean value of the $N = 3$ independent repeats and the error bars show ±1 standard deviation. **D** Violin plots showing the distribution of encapsulated protein concentrations versus the diameter of the GUVs. GUV diameters were binned so each integer value includes GUVs with diameters ±0.5 μm of the integer value. A Kruskal-Wallis (KW) ANOVA test shows that the encapsulated protein concentrations in the lumen is independent of the diameter of the GUVs ($X^2(6, 617) = 2.01$, $p = 0.92$). The data shown is for a loading concentration of 1.75 μM FITC-BSA. The horizontal black and red lines are the mean and median values of the distributions respectively.

that range from 0.25 to 0.4 (Fig. 2B)[12,13,17,18]. The mean encapsulated concentration of FITC-BSA had a 1:1 correlation with loading concentration (Fig. 2C) regardless of GUV size (Fig. 2D). While there were some empty GUVs after loading, the fraction did not vary with protein concentrations and was removed from the analysis.

### Reconstitution of PTOs in GUVs

PTO proteins were encapsulated in GUVs using the PAPYRUS-wDL method and assumed to follow the same encapsulation statistics as FITC-BSA to create femtoliter volume PTOs partitioned into GUVs. We term these GUVs that encapsulate the PTO proteins in their lumens as PTO-GUVs. Under standard in vitro bulk-solution conditions, the monomeric concentrations of PTO proteins are 1.2 μM, 3.5 μM, and 3.5 μM of KaiA, KaiB, and KaiC, respectively[19]. We refer to this concentration as the 1× PTO concentration. Cyanobacteria have PTO concentrations around 2.1×[4,19], thus, significantly higher than the concentration required for robust oscillations under bulk solution conditions. The apparent high expression levels of PTO proteins in cyanobacteria compared to what is necessary to obtain robust oscillations in vitro have not been adequately explained in the literature. We separately prepared PTO-GUVs with mean PTO concentrations of 0.5×, 0.75×, 1.0×, 1.5×, and 2.5× where 50% of KaiB was composed of a construct labeled with the thiol-reactive fluorophore 6-iodoacetamidofluorescein (6IAF)[14]. The fluorescence intensity of KaiB-6IAF is quenched upon binding KaiC (Supplementary Fig. 1).

Thus, in oscillating PTO-GUVs, the fluorescence intensity is high during the subjective day when KaiB is not bound to KaiC and low during the subjective night when KaiB binds KaiC to form the KaiABC nighttime complex[19–21]. By measuring fluorescence from KaiB-6IAF using confocal microscopy, we followed PTO oscillations over several days at 30 °C in single PTO-GUVs with diameters of 2.0 ± 0.5 μm, 3.0 ± 0.5 μm, 4.0 ± 0.5 μm, 6.0 ± 0.5 μm, 8.0 ± 0.5 μm, and 10.0 ± 0.5 μm. The 2 μm PTO-GUVs corresponds to the average volume of wildtype *S. elongatus*[4]. We show in Fig. 3A a typical image of the PTO-GUVs prepared to have a mean PTO concentration of 1×. Time-lapse confocal images for three PTO-GUVs labeled V1, V2, and V3 are shown in Fig. 3B–D. The mean fluorescence intensity of V1 and V2 oscillated with a 23-h period while the fluorescence intensity of V3 remained constant. A single-sided amplitude spectrum showed a clear peak corresponding to 23 h for V1 and V2 and no peak for V3 (see "Methods" for details of FFT analysis). We show additional confocal images of PTO-GUVs with sizes ranging from 2 μm to 10 μm and the time traces of their fluorescence intensities in Supplementary Fig. 2.

### Fidelity depended on PTO loading concentration and GUV size

Next, we compared the oscillation of PTOs in bulk solution (Fig. 4A) and in PTO-GUVs (Fig. 4B). Consistent with previous results[8,19,22], the bulk reactions were robust up to 0.75×, where robust is defined as sustained oscillations that persist for 100 h. In comparison, the robustness of the partitioned PTOs averaged over the 2.0 μm GUVs was

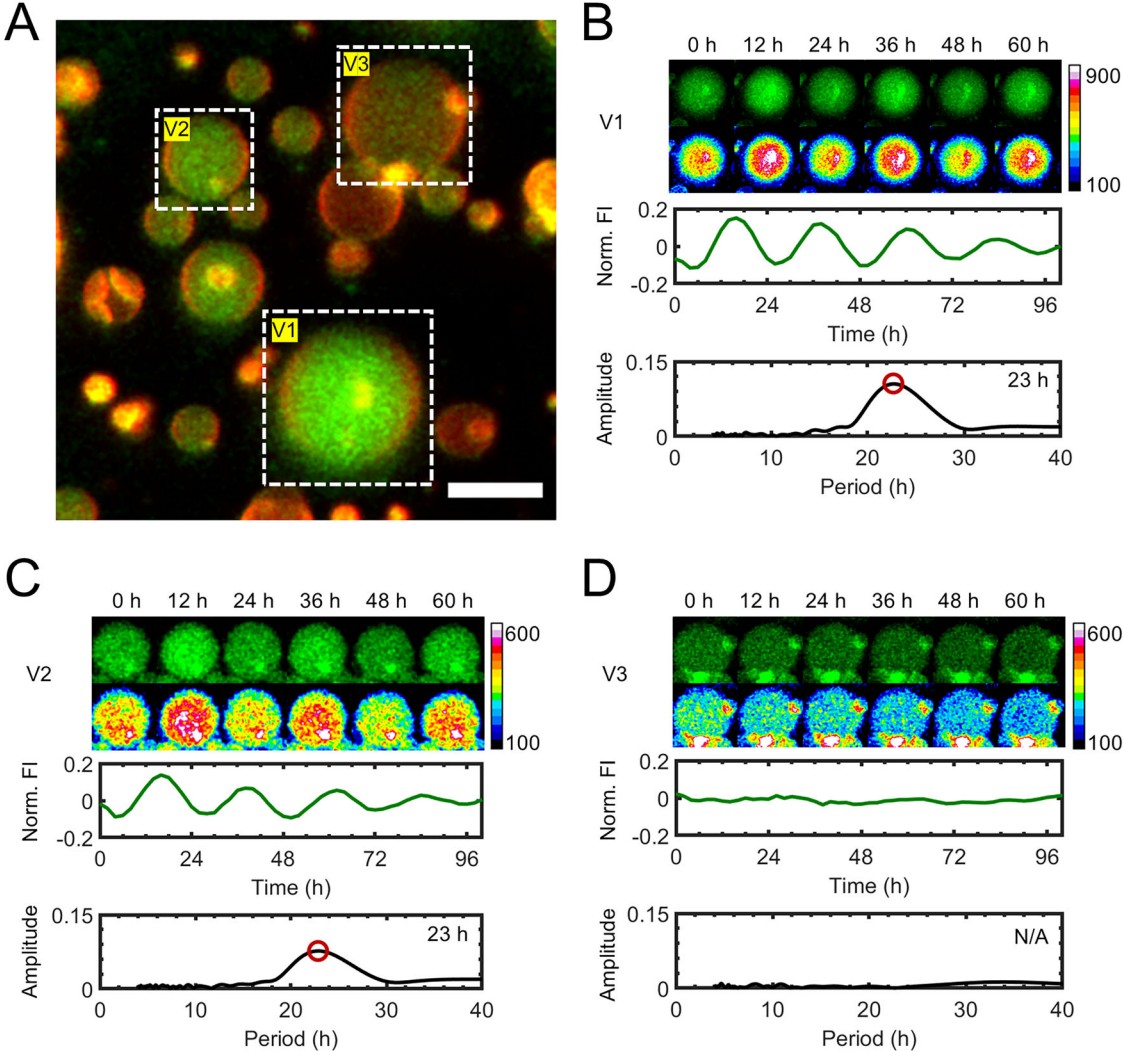

**Fig. 3 | PTO-GUVs show circadian oscillations, though some do not oscillate.**
**A** A representative confocal image of PTO-GUVs prepared with a loading concentration of 1× PTO. The fluorescently-labeled GUV membranes are false-colored red and the KaiB-6IAF in the lumens are false-colored green. Scale bar 10 μm.
**B**–**D** Top images: still images of the PTO-GUVs labeled V1 through V3 (green) at 12-h intervals. Bottom images: intensity color-mapped images to emphasize the changes in mean fluorescence intensity in the lumens. The color mapping is linear, and the range was adjusted to the maximum intensity of each image. Top plots show the time traces of the normalized mean fluorescence intensity (Norm. FI) for each PTO-GUV. The bottom plots are the single-sided amplitude spectrum of the respective traces. The position of the peak in the amplitude spectrum in (**B**) and (**C**) are marked with an open-faced red circle. The inset text is the period corresponding to the peak. The sample temperature was maintained at 30 °C. We show additional representative images at the other mean PTO concentrations in Supplementary Fig. 2. Experiments were repeated at least two times with similar results.

significantly lower. Only the cell-like 2.5× concentration showed detectable population-averaged oscillations. All other concentrations had no detectable population-level oscillations (Fig. 4B).

Clock fidelity monotonically increased with loading concentration across all sizes of GUVs (Fig. 4C), suggesting that higher mean PTO concentrations can buffer against the effect of variation in protein concentrations. Recall that the 1× PTO concentration is the standard for in vitro bulk solution studies. However, in cell-size 2 μm PTO-GUVs, the fidelity was only 7 % at 1× PTO concentrations but increased to 70 % at 2.5× PTO concentrations. Thus, we conclude that the lack of population-level oscillations in the partitioned PTOs is due to individual PTO-GUVs failing to oscillate.

The fidelity of the PTO-GUVs increased with GUV size for all PTO concentrations, suggesting a size effect. Calculation of the monomeric copy numbers of PTO proteins, which depend on the volume and concentration of the PTO proteins, showed that the copy numbers of all proteins in the PTO-GUVs were >1500. Thus, molecular noise is not

expected to have a significant effect on reaction rates. Screening other size parameters, we find that the fidelity data points collapsed to straight lines with negative slopes when potted against the surface area-to-volume (SA/V) ratio (Fig. 4D), hinting at the importance of the membrane.

Histograms aggregating the periods of oscillation show that the mean of the periods was similar within 1 h and thus appear to be independent of mean PTO concentration and GUV size. The upper and lower range of the periods spans 4 h (Fig. 4E). The mean amplitudes of the oscillations, which correlate with the fraction of KaiC that participates in the clock reaction[7] decreased with decreasing PTO concentration (Fig. 4F).

This negative slope between clock fidelity and SA/V ratio (Fig. 4D) suggested that the membrane is playing a negative role in the fidelity of the PTO. Confocal images of GUVs encapsulating KaiB-6IAF show evidence of KaiB binding to the membrane. This observation is consistent with observations in *S. elongatus*, which show a significant

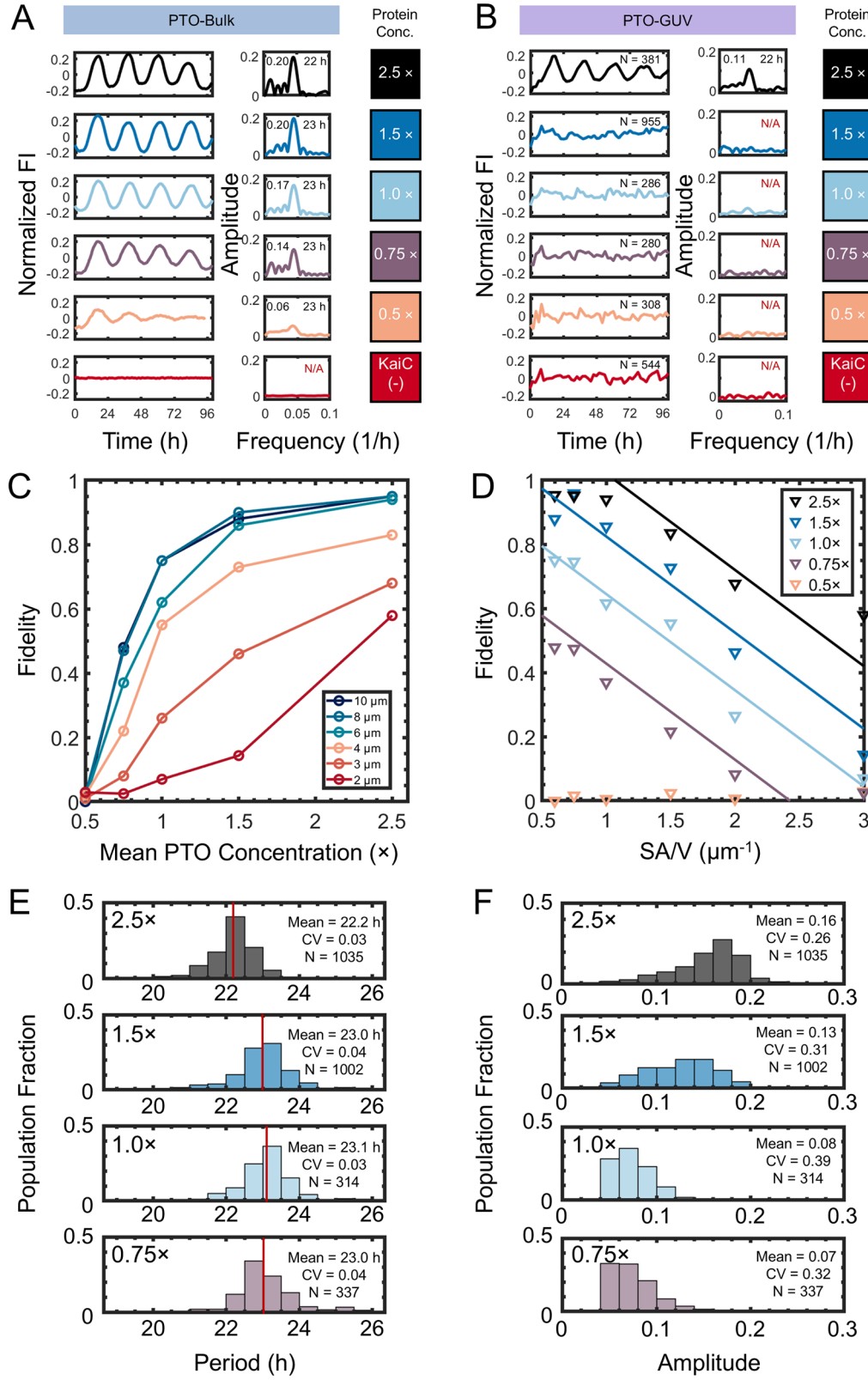

fraction of KaiB localizing to membranes depending on the time of day[23]. In Fig. 5A a bright ring can be seen in the KaiB-6IAF channel (green) in the region where the GUV membrane (red) is located, and a strip intensity profile shows a spike in KaiB-6IAF intensity (green) at the membrane (gray) as indicated by the black arrows. Figure 5B demonstrates that FITC-BSA does not bind to GUV membranes even though both BSA and KaiB were conjugated to fluorescein derivatives. These

images and the corresponding strip intensity profiles suggest that KaiB and not 6IAF is responsible for membrane binding and rules out the possibility of bleed-through from the rhodamine channel.

### Modeling the PTO-GUVs

Coupled deterministic[24–26] or stochastic[4,27] differential equations of simplified partial reactions of the multimeric PTO proteins have been

**Fig. 4 | The PTO in GUVs show different behavior from the PTO in the bulk.**
**A** Bulk in vitro (PTO-Bulk) measurements of the fluorescence intensity (FI) of KaiB-6IAF across a range of mean post-translational oscillator (PTO) concentrations (0.5× to 2.5×, various colors) evaluated using 50 μL reaction volumes in a fluorescence plate reader. The normalized time trace of the FI (detrended and normalized by the mean intensity) over time is shown in the left panel and the amplitude spectrum from the fast Fourier transform (FFT) analysis is shown in the right panel. The text shows the amplitude (top left) and period (top right) obtained from the global peak of the amplitude spectrum. **B** Population-averaged KaiB-6IAF FI from PTO-GUVs 2 ± 0.5 μm in diameter for varying mean PTO concentrations (0.5× to 2.5×, various colors). The population-averaged and normalized PTO amplitude (detrended and normalized by mean intensity) over time is shown in the left panel.

The inset text reports $N$ = the total number of PTO-GUVs for each concentration. The amplitude spectrum is shown in the right panel. **C** Plot of the PTO fidelity versus the mean PTO concentration (0.5× to 2.5×) across varying PTO-GUV sizes (Ø: 2 μm to 10 μm). **D** Plot of the fidelity versus the surface area to volume ratio (SA/V) across varying PTO-GUV sizes (Ø: 2 μm to 10 μm). The straight lines are guides for the eye. **E** Histograms of the periods of oscillating PTO-GUVs. The red line shows the mean period for each distribution. The text inset reports the mean period, the coefficient of variation (CV) of the period, and $N$ = the number of PTO-GUVs analyzed. **F** Histograms of the amplitude of oscillating PTOs. The text inset reports the mean amplitude, the coefficient of variation (CV) of the amplitudes, and $N$ = the number of PTO-GUVs analyzed. The histogram in (**E**) and (**F**) includes PTO-GUVs of all diameters.

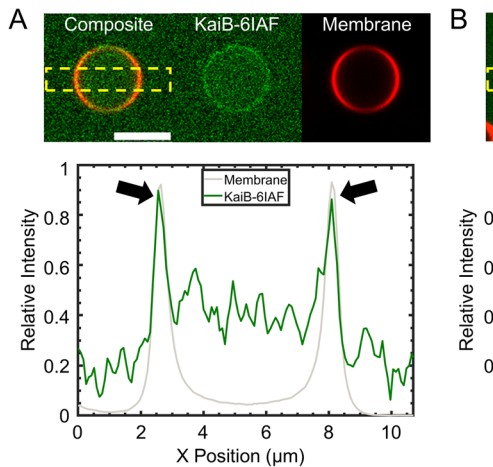
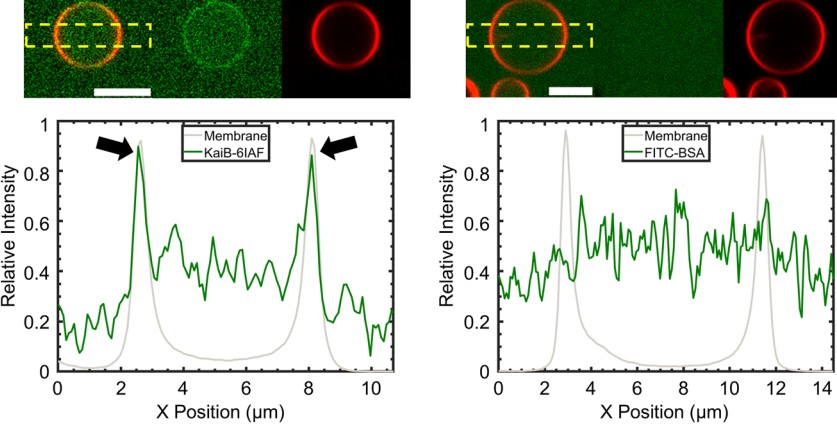

**Fig. 5 | KaiB binds to GUV membranes.** Multichannel confocal images of GUVs encapsulating KaiB-6IAF or FITC-BSA. Images of the membrane channel are false-colored red and images of the protein channel are false-colored green. The leftmost panels show a composite image of the two channels, the middle panels show the protein channel, and the rightmost panels show the membrane channel. The lower plots are strip-intensity profiles of the regions shown in the rectangular boxes with the dashed yellow lines. **A** For the KaiB-6IAF sample, the images show a localized

bright ring around the membrane in the protein channel. The strip intensity profile shows a colocalized region of high KaiB-6IAF intensity (green line) at the membrane (gray line). The colocalized region is highlighted by the thick black arrows. **B** There is no corresponding bright ring in the FITC-BSA channel or region of high FITC-BSA intensity at the membrane. Experiments were repeated at least two times with similar results. Scale bars = 5 μm.

used to model PTO oscillations. Here we develop an experimentally guided phenomenological mathematical model to describe the effect of the GUV size and PTO concentration on the oscillations in PTO-GUVs. The distribution of encapsulated protein concentrations followed a gamma distribution that did not vary with GUV size (Fig. 2). Our data also demonstrated that the mean concentrations of encapsulated proteins equaled the loading concentrations with cell-like variation (CV-0.3). We modeled the PTO-GUVs by simulating 5000 GUVs of sizes from 2 μm to 10 μm for the four loading concentrations (30,000 total GUVs). Each PTO-GUV contains some concentrations of KaiA, KaiB, KaiC, and KaiABC complexes independently and randomly assigned by sampling from a corresponding gamma distribution centered around the mean concentration of each PTO protein. The free concentration of KaiB is calculated to include membrane binding and depends on the diameter of the GUVs (see "Methods" for details). We assume that membrane-bound KaiB does not participate in PTO reactions. This assumption is supported by the apparent negative role that the membrane plays in the fidelity of the PTO (see Fig. 4D). Figure 6A shows a three-dimensional scatter plot representation of the concentrations in the simulated PTO-GUVs. The axes are the concentrations of KaiA, KaiB, and KaiC. The lowest PTO concentration (0.5×) results in a cloud of points at the lower bottom right of the plot (light yellow). With increasing mean PTO concentrations, the cloud of points shifts to the upper left, and the range of concentrations, i.e., the width of the cloud, increases (darker colors).

The PTO-GUVs are categorized as either "oscillating" or "non-oscillating" based on established thresholds for minimum concentrations and stoichiometric ratios of the PTO in bulk solutions. Oscillating PTO-GUVs for all loading concentrations and diameters are shown as green dots (Fig. 6A). The model shows that oscillating PTO-GUVs occupy the left half of the cloud. There are higher numbers of oscillating PTO-GUVs at higher mean PTO concentrations and lower numbers of oscillating PTO-GUVs at lower mean PTO concentrations. There are almost no oscillating PTO-GUV at the lowest mean PTO concentration. A two-dimensional representation of the model shows excellent correspondence of the fidelity of the simulated PTO-GUVs with the experimental PTO-GUVs (Fig. 6B). We also calculate the periods and amplitudes of each oscillating PTO-GUV using experimentally derived relationships between protein concentration and stoichiometry and plot the corresponding histograms in Fig. 6C, D. We provide details of the calculations in the "Methods". The distribution of periods and amplitudes obtained from the model follows the distribution of experimental PTO-GUV periods and amplitudes well (compare Fig. 6C, D with Fig. 4E, F).

### Insights into the fidelity and synchrony of the PTO in cyanobacteria

Since our phenomenological model can reproduce the fidelity, periods, and amplitudes of the PTO-GUVs, we extend the model to obtain insights into PTO function in cyanobacteria. On average, 50% of KaiB is

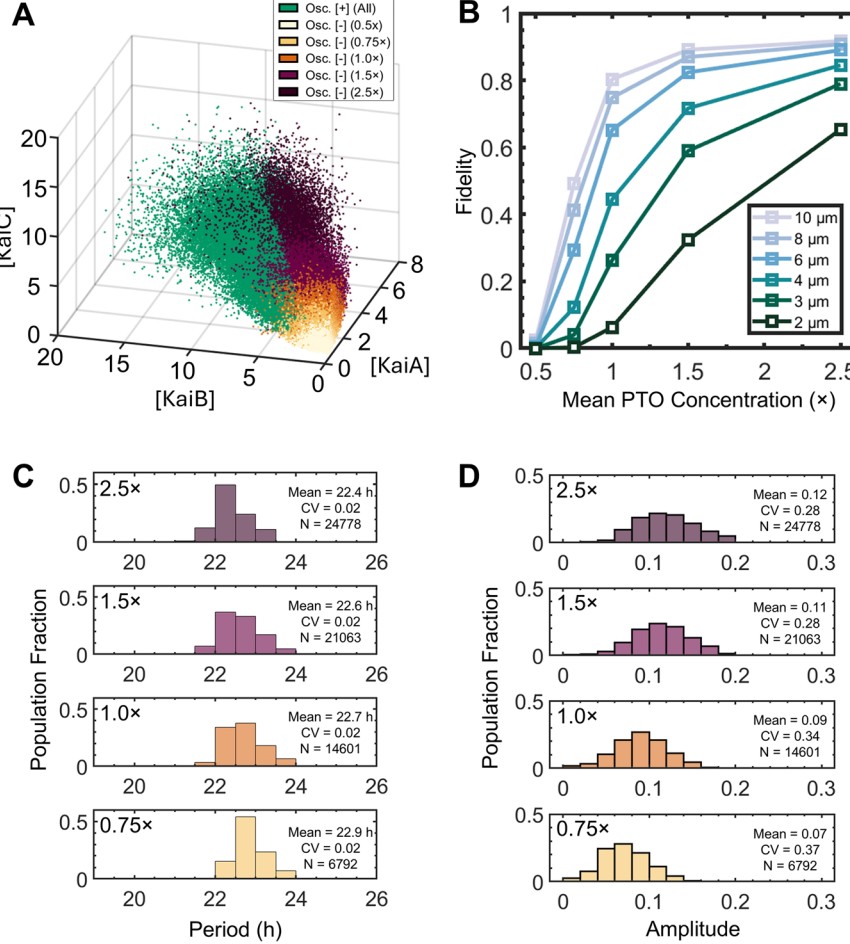

**Fig. 6 | Modeling of the PTO-GUVs. A** 3D scatter plot of the distribution of KaiA, KaiB, and KaiC protein concentrations (monomeric) in the simulated PTO-GUVs. The plot aggregates a range of clock protein concentrations (0.5× to 2.5×, various colors) and a range of GUV sizes (2 μm, 3 μm, 4 μm, 6 μm, 8 μm, and 10 μm, all ±0.5 μm). PTO-GUVs that were determined to be oscillating (Osc. [+]) are colored green. All non-oscillating GUVs (Osc. [-]) are color-coded by concentration as indicated in the legend. **B** Plot of the fidelity of simulated PTO-GUVs obtained from the model. **C** Histogram of the periods of the simulated oscillating PTO-GUVs obtained from the model. The text inset shows the mean period, the coefficient of variation (CV) of the period, and *N* = the number of oscillating simulated PTO-GUVs. **D** Histogram of the amplitudes of the simulated oscillating PTO-GUVs obtained from the model. The text inset reports the mean amplitude, the coefficient of variation (CV) of the amplitudes, and *N* = the number of oscillating simulated PTO-GUVs from the model. The histograms in (**C**) and (**D**) include PTO-GUVs of all diameters.

associated with the membrane in cyanobacteria[23]. Other components of the *S. elongatus* circadian clock, such as SasA[28] and CikA[29], interact with PTO proteins and have been shown to effectively ease the conditions required for the PTO to oscillate. For example, the presence of 1.0 μM SasA allows PTO rhythms to occur even if KaiB drops below a concentration of 1.8 μM[9]. Similarly, the presence of 0.9 μM of CikA allows the PTO to oscillate down to a KaiA concentration of 0.3 μM[8,30]. Figure 7A shows the predicted fidelity in simulated cyanobacteria without SasA and CikA (PTO) and with SasA and CikA (PTO'). Without CikA and SasA, the fidelity of the PTO across the population was only 85.9% at cellular conditions (2.1×). With CikA and SasA, the fidelity increases to 99.6 %. A similar level of fidelity cannot be obtained without SasA and CikA even if the mean PTO concentration was increased to 10× because the fidelity plateaued at 86.3%. This result suggests the importance of these "accessory" proteins for the robust oscillation of the PTO in a clonal population, which was not apparent in bulk in vitro experiments.

With SasA and CikA, the fidelity reaches a plateau of 99.6 % at 1.5×. In the cyanobacteria, KaiC protein levels change by ~50 % over the course of the cell's circadian cycle, and KaiB protein concentrations change by ~60%[23]. Expressing the PTO proteins at 2.1× can help ensure that the PTO concentration does not reach the region where

there is a steep decrease in PTO fidelity as the protein concentration changes over the course of the cell's circadian cycle. We suggest that this steep drop-off in fidelity at mean PTO concentrations below 1.5× explains why cyanobacteria express the PTO proteins at a relatively high level.

Along with high fidelity, cyanobacteria also maintain a high level of synchrony between clonal cells at constant conditions[3,11]. Although the variation in periods between the PTOs due to variation in protein concentrations is relatively small, over time, these small variations in period could lead to large shifts in phase and the eventual loss of synchrony between the PTOs. We extend our model to include time-dependent variation in protein concentrations to understand the remarkable synchrony of clock-controlled gene expression in cyanobacteria. Our model assumes that the bacteria undergo cell division every 24 h. We model three scenarios, 1) the bacteria maintain the protein concentration that was assigned at the beginning of the simulation according to the gamma distribution with a CV of 0.25 at each cell division, i.e., perfect memory, 2) the protein concentration in the bacteria randomly changes according to the rules of the gamma distribution with a CV of 0.25 at each cell division, i.e., no memory, and 3) the protein concentration in the bacteria randomly changes with each cell division according to the rules of the gamma distribution with

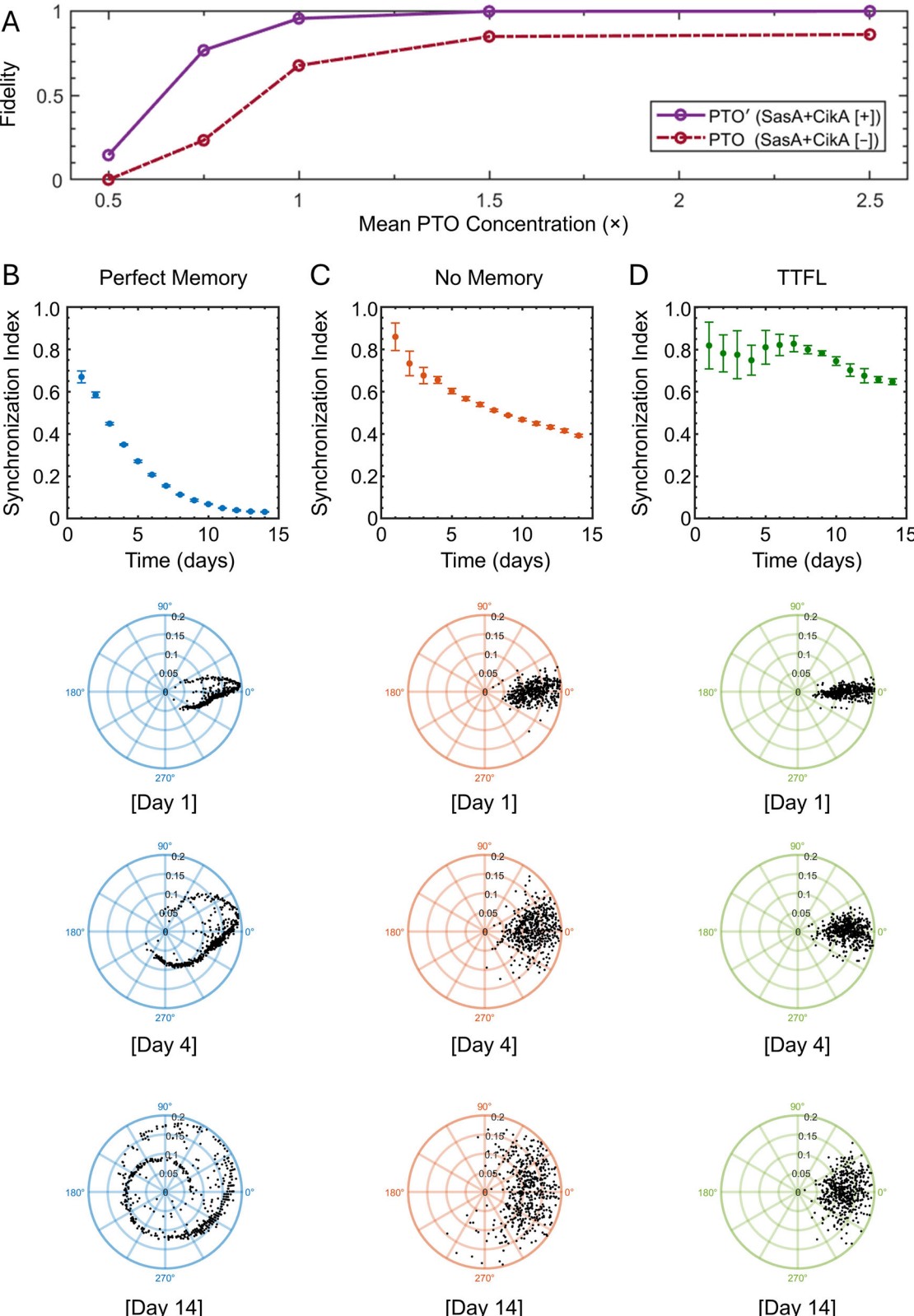

**Fig. 7 | Insights into PTO fidelity and synchrony under constant conditions.** **A** Prediction of the fidelity as a function of mean PTO concentration with SasA and CikA (PTO′ (SasA+CikA[+]), solid purple line) and without SasA and CikA (PTO (SasA+CikA[−]), dashed red line) in simulated cyanobacteria. **B**–**D** Plots of the synchronization index and polar plots of the phase (angular axis) and the amplitude (radial axis) for the perfect memory, no memory, and perfect memory + TTFL models. The synchronization index is plotted for 14 days and was calculated from the time traces of 5000 simulated cyanobacteria. The data are presented as the mean values ± 1 standard deviation. The upper polar plots show Day 1, middle polar plots show Day 4, and lower polar plots show Day 14. For clarity, each polar plot contains $N = 500$ data points randomly selected from the 5000 simulated cyanobacteria.

a CV of 0.25 at each cell division but the TTFL actively corrects the concentrations by the end of each cell cycle to an average value, i.e., no memory + TTFL. We calculated the circadian traces of 5000 bacteria for 14 days. We then calculated the synchronization index (SI) of the phase shifts over the course of 14 days. Scatter plots of the SI and polar plots of the evolution of the phase angle with time[11,31] are shown in Fig. 7B-D. An SI of 1 indicates perfect synchrony while an SI of 0 shows no synchrony. Tightly clustered points on the polar plots indicate a high degree of synchrony of the phases.

In the perfect memory model, which is akin to the PTO-GUVs in our experiments, the synchronization index drops to 0.35 at 4 days and then 0.03 after 14 days (Fig. 7B). By day 14, the phase angles become completely dispersed across the polar plot.

In the no memory model, the SI was 0.66 at 4 days and 0.39 after 14 days (Fig. 7C). Thus, we find that noise due to variation in protein concentration at regular cycles, instead of leading to loss in synchrony, promotes synchronization by limiting the duration of adverse stoichiometries on the period. Despite this increase in synchrony relative to the perfect memory scenario, the phase angles show a relatively wide dispersion at Day 4, unlike what is observed in WT cyanobacteria[11]. The synchronization index and polar plots of the phase shift of PTO-controlled YFP expression in cyanobacteria that have only their PTO circuit and their TTFL abrogated[11] show a behavior that is intermediate to the perfect memory and no memory model, suggesting that cyanobacteria have some memory of their protein concentration after cell division.

Including a simplified TTFL correction mechanism results in the recapitulation of the behavior observed for WT cyanobacteria. After 4 days, the synchronization index was 0.75, and after 14 days the SI was 0.65 (Fig. 7D). Importantly, the phase angles remain tightly clustered on the polar plots similar to what is observed for WT cyanobacteria[11].

We conclude that high expression levels of the PTO proteins along with SasA and CikA expression are required to maintain high fidelity of the PTO in cyanobacteria and that the TTFL is necessary to maintain phase synchrony of the PTO across the population.

## Discussion

The circadian clock of *S. elongatus* has near-perfect fidelity and phase synchrony despite their tiny volumes. We explored possible contributions to this fidelity and synchrony by investigating PTOs reconstituted in GUVs. Studies in GUVs avoid the myriad hard-to-control cellular processes inherent with any in vivo approach while allowing close mimicking of cellular conditions compared to previous bulk in vitro approaches. Advantageously, the PAPYRUS-wDL method encapsulates proteins with cell-like distributions independent of loading concentration and GUV size. We could measure oscillations in PTO-GUVs with diameters down to 2 μm (4 fL) and up to 10 μm (~500 fL) to determine effects in cell-like volumes as well as in volumes beyond what is available in vivo.

The strong dependence of fidelity on PTO concentration suggests that *S. elongatus* maintains high levels of PTO proteins to suppress clock dysfunction arising from noise. Binding to membranes reduces free concentrations, thus requiring higher PTO expression levels. Indeed, in dividing *S. elongatus* cells engineered to have tunable PTO expression, it was found that decreasing PTO levels below 2.1× reduced phase coherence of clock-controlled YFP expression between cells and that traces of YFP intensities show widely varying non-circadian periods[4]. Our results show that the PTOs in GUVs are constrained to circadian periods even under large dilutions in protein concentrations. When the PTO protein concentrations are too low, the PTOs stop oscillating. We suggest that the large variation in periods observed in vivo of PTO-controlled gene expression can be explained by the PTO ceasing to function in an increasing fraction of the cells. When the PTOs cease to oscillate, gene expression is no longer clock-controlled

until protein concentrations are restored, which should result in large variations in apparent periods.

If intracellular levels of gene-regulatory proteins are highly stochastic, as happens when the levels of those proteins are low enough, there can be significant cell-to-cell variability within an isogenic population[12]. *S. elongatus* appears to address this problem by expressing the Kai proteins at relatively high levels along with accessory proteins, SasA and CikA. Other cyanobacteria, such as the multicellular and filamentous cyanobacterium, *Anabaena* sp. PCC 7120, appears to employ intercellular communication that couples PTOs across the cells of a filament[18] to generate robust and phase-coherent circadian rhythms with significantly lower PTO levels than *S. elongatus*[32]. Intercellular coupling of clocks is reminiscent of coupling of clocks in higher multicellular organisms where clock proteins occur at nanomolar concentrations[33].

Our model reached 99% fidelity for GUVs with PTO concentrations (2.1×) and cell-like diameters (2 ± 0.5 μm) by accounting for the buffering effects of CikA and SasA. There is indirect evidence supporting this notion of buffering in vivo. For example, deleting the *sasA* gene lowered the expression of KaiB and KaiC and cells were virtually arrhythmic[28]. Restoring KaiB and KaiC expression to normal levels in a *sasA* knockout strain only partially restored bioluminescence rhythms[8]. We also show that the TTFL is essential for maintaining phase synchrony between bacteria. The implication is that the TTFL corrects for differences in stoichiometry that will result in large phase shifts if not corrected. Looking ahead, our approach of using relationships of the reaction parameters of proteins measured in the bulk to explain partitioned reactions in cell-like volumes could be applicable to other cellular processes that involve the collective behavior of large numbers of proteins such as liquid-liquid phase separation[34,35]. Furthermore, using similar techniques to incorporate the full clock with DNA binding[7,36] would be an important step toward building a synthetic cell that can show circadian control of gene expression.

## Methods
### Materials
We purchased Gold Seal™ 60 mm × 22 mm glass coverslips, Fisherbrand ™ Premium Plain Glass Microscope Slides (75 mm × 25 mm), CELLSTAR® black clear bottom 96 well plates (Greiner), Coplin glass staining jars (DWK Life Sciences), Corning® 15 mm diameter regenerated cellulose syringe filters (0.2 μm pore size), and MilliporeSigma™ Ultrafree™-MC centrifugal filter devices (0.22 μm pore size) from Thermo Fisher Scientific (Waltham, MA). We purchased artist grade tracing paper (Jack Richeson & Co., Inc.), circular hole punches (EK Tools Circle Punch, 3/8 in.), square hollow punch cutters (Amon Tech) from Amazon Inc. (Seattle, WA).

### Chemicals
We purchased sucrose (BioXtra grade, purity ≥99.5%), glucose (BioXtra grade, purity ≥99.5%), bovine albumin-fluorescein isothiocyanate conjugate (FITC-BSA) (albumin from bovine, ≥7 mol FITC/mol albumin), sodium chloride (NaCl) (ACS grade, VWR International), magnesium chloride hexahydrate ($MgCl_2 \cdot 6H_2O$) (ReagentPlus grade, purity ≥99%, Sigma-Aldrich), and ethylenediaminetetraacetic acid (EDTA) (BioReagent grade, purity ≥98.5%) from Sigma-Aldrich (St. Louis, MO). We purchased chloroform (ACS grade, purity ≥99.8%, with 0.75% ethanol as preservative), 1 N potassium hydroxide (KOH) (Certified grade, 0.995 to 1.005 N, Fisher Chemical), 3-aminopropyl trimethoxysilane (APTES) (purity ≥98.5%, ACROS Organics), glacial acetic acid (ACS grade, purity ≥99.7%, Fisher Chemical), methanol (ACS grade, purity ≥99.8%, Fisher Chemical), adenosine 5′-triphosphate (ATP solution) (Tris-buffered, purity >99% via HPLC, Thermo Scientific) from Thermo Fisher Scientific (Waltham, MA). We obtained 18.2

MΩ·cm Type I ultrapure water from an ELGA Pure-lab Ultra water purification system (Woodridge, IL).We purchased 1,2-dioleoyl-*sn*-glycero-3-phosphocholine (DOPC), 1,2-distearoyl-*sn*-glycero-3-phos-phoethanolamine-N-(methoxy(polyethylene glycol)-2000) (PEG2000-DSPE), 1,2-distearoyl-*sn*-glycero-3-phosphoethanolamine-N-(biotinyl(polyethylene glycol)-2000) (PEG2000-DSPE-Biotin), and 1,2-dioleoyl-*sn*-glycero-3-phosphoethanolamine-N-(lissamine rhodamine B sulfonyl) (Rhod-DOPE) from Avanti Polar Lipids, Inc. (Alabaster, AL). We purchased NHS-ester polyethylene glycol (PEG) (5 kDa) and biotinylated NHS-ester PEG (biotin-PEG) (5 kDa) from Laysan Bio, Inc. (Arab, AL).

### Kai clock protein expression, purification, and labeling

Freshly transformed cells harboring the expression construct (seKaiA-1-284, seKaiB-1-102-FLAG, sekaiB-1-102-K25C-FLAG, FLAG-seKaiC-1-519) were used for the inoculation of a starter culture in lysogeny broth (LB) medium with 50 mg/mL kanamycin sulfate and incubated in a shaker at 37 °C and 220 rpm for 6.5 h[14]. We then transferred 5 mL of the starter culture to 1 L of M9 medium with 0.2% D-glucose, 2 mM $MgSO_4$, 0.1 mM $CaCl_2$, and 50 μg/mL kanamycin sulfate. The cells grew at 37 °C and 220 rpm until an optical density of 0.6 at 600 nm was reached. We induced protein expression by introducing 0.2 mM isopropyl β-d-1-thiogalactopyranoside to the cells and further incubated at 30 °C in the shaker for 12 h.

We harvested the cells and used an Avestin C3 Emulsiflex homogenizer to lyse the cells. The cell lysate was spun down by centrifugation at 27,000 × *g* for 45 min at 4 °C. We then performed affinity purification with Ni-NTA columns at 4 °C using lysis buffer (50 mM $NaH_2PO_4$, 500 mM NaCl, pH 8.0), wash buffer (50 mM $NaH_2PO_4$, 500 mM NaCl, 20 mM imidazole, pH 8.0), and elution buffer (50 mM $NaH_2PO_4$, 500 mM NaCl, 250 mM imidazole, pH 8.0). Then we added ULP1 for the cleavage of the 6×His-SUMO fusion protein at 4 °C for 15 h. We loaded the cleaved protein on a Ni-NTA column again to remove the 6×His-SUMO. We concentrated the flow-through with a 10 kDa molecular weight cut-off (MWCO) membrane filter in an Amicon™ stir-cell concentrator at 4 °C, then further purified the recovered protein by gel-filtration chromatography[14].

For fluorescent labeling of KaiB, we mixed the purified KaiB-K25C with 80 μL of 12.5 mg/mL 6-(Iodoacetamido)fluorescein (6-IAF) suspended in methanol and incubated at 4 °C for 15 h. The sample was then concentrated to 2 mL with a 10 kDa MWCO membrane filter in an Amicon stirred cell concentrator, then further purified by gel-filtration chromatography to separate labeled protein from free fluorophore[14].

### Preparation of biotin-PEG functionalized glass

Biotin-PEG functionalized glass allows vesicles to be bound to the glass surface through streptavidin-biotin interactions, effectively preventing lateral and axial diffusion of the vesicles during imaging. The lipid mixture used to assemble vesicles must contain a biotinylated lipid for this methodology to work. The procedure to prepare the functionalized slides outlined below is primarily based on procedures reported in ref. 37 with some modifications.

First, we scratched ten 60 × 22 mm glass coverslips and ten 75 × 25 mm glass slides with a small line in the top right corner using a diamond-tipped scriber. The line allowed identification of which surface is functionalized. Otherwise, both the functionalized and non-functionalized surfaces appear identical. Working in a chemical safety hood, we placed the ten marked glass coverslips and ten marked glass slides together into a Coplin glass staining jar with each of the five slots containing one glass coverslip and one glass slide. The glass slides and coverslips were placed so that the marked surfaces to be functionalized were facing away from one another. The jar containing the glass was filled with 50 mL of ultrapure water, swirled, and then emptied.

These steps were repeated two more times. The jar was filled with 50 mL of acetone and placed in a bath sonicator in a chemical safety

hood for 20 min. We then discarded the acetone and rinsed the jar with ultrapure water three times. The jar was next filled with 50 mL of 1 N KOH and placed in a bath sonicator for 30 minutes. We then left the jar in the chemical safety hood overnight to allow KOH to etch the surface layer of the glass.

The next day, we prepared a 50 mL "silanization mixture" of 3-aminopropyl trimethoxysilane (APTES) and acetic acid at 10 v/v% and 5 v/v% in methanol, respectively. We then discarded the KOH and rinsed the jar with ultrapure water three times. The silanization mixture was then added to the jar and incubated for 30 min. After 30 min, we discarded the silanization mixture in an appropriate waste container and rinsed the jar with 50 mL of neat methanol 3 times. Then, we dried the coverslips using a stream of ultrapure nitrogen gas from a nitrogen gun.

We filled empty 10–100 μL or 100–1000 μL plastic pipette tip boxes, with the insert included, with 20 mL of ultrapure water to act as humidity chambers (Supplementary Fig. 3A). We prepared a mixture of NHS-PEG (5 kDa) and biotinylated NHS-ester PEG (5 kDa) at a concentration of 125 mg/mL and 3.1 mg/mL, respectively, in 0.1 M sodium bicarbonate buffer (pH 8.5). 64 μL of this mixture was sandwiched between two sets of marked surfaces of the coverslips or slides. The surfaces were placed into the humidity chamber, propped up by pipette tips (Supplementary Fig. 3B). Any air bubbles between the glass surfaces were removed by applying gentle pressure. We closed the lid of the pipette tip boxes and placed the box in a benchtop cabinet that was cool and protected from light. Functionalization was allowed to continue overnight. The next day, we carefully took the glass sandwiches apart and rinsed the surfaces with ultrapure water. We then dried the slides using a stream of ultrapure nitrogen gas from a nitrogen gun. Each pair of glass slides and coverslips was stored in a 50 mL Falcon™ conical centrifuge tube in a −20 °C freezer such that the functionalized sides were not in contact with each other or the walls of the centrifuge tubes.

### Buffers

10× clock buffer consisted of 200 mM Tris, 1500 mM NaCl, 50 mM $MgCl_2$, 10 mM ATP, and 5 mM EDTA, and was based on the buffer optimized for cyanobacterial circadian clock proteins when diluted to 1×[14]. The initial budding buffer consisted of 119 mM sucrose. The final buffer composition after GUV assembly consisted of 1× clock buffer and 100 mM sucrose. The sedimentation buffer was equimolar with the budding buffer, consisting of 100 mM glucose and 1× clock buffer. All buffers and solutions were filtered through a 0.2 μm regenerated cellulose syringe filter.

### Phospholipid mixtures

The standard phospholipid mixture used in studies with GUVs consisted of a 1 mg/mL solution of DOPC:PEG2000-DSPE:PEG2000-DSPE-Biotin:Rhod-DOPE at 94.4:5.0:0.5:0.1 mol% in chloroform. Here, DOPC (94.4 mol %) was the primary bilayer forming phospholipid. PEG2000-DSPE (5.0 mol %) is a PEG functionalized lipid that provides steric repulsion and has the primary role of inhibiting aggregation of vesicles in the salty clock buffer. PEG2000-DSPE-Biotin (0.5 mol %) is a PEG functionalized lipid with a terminal biotin moiety that allows streptavidin-biotin binding interactions used to immobilize GUVs to the biotinylated glass surfaces. Rhod-DOPE is a rhodamine functionalized lipid that allows the visualization of bilayer membranes.

### Protein solutions

Protein solutions were prepared at 15× the intended final concentrations in 1× clock buffer and filtered through a 0.2 μm regenerated cellulose syringe filter. FITC-BSA was used as the model protein for investigating encapsulation statistics. 1.0× PTO reactions contained 1.2 μM KaiA, 1.75 μM KaiB, 1.75 μM KaiB-6IAF, and 3.5 μM KaiC. We scaled the other PTO concentrations accordingly.

## Assembly of giant vesicles using PAPYRUS with diffusive loading (PAPYRUS-wdL)

We deposited 10 μL of a 1 mg/mL lipid solution onto a 9.5 mm diameter circular cutout of tracing paper using a Hamilton glass syringe[15,16]. The lipid was dispensed slowly, nearly parallel to the paper, while simultaneously using the long edge of the syringe tip to evenly spread the lipid across the tracing paper as the chloroform evaporated. During this process, the tracing paper was held by clean metal tweezers and not placed down until the chloroform had completely evaporated. We then placed the lipid-coated tracing paper into a standard laboratory vacuum chamber for 1 h to remove traces of residual chloroform.

Diffusive loading allows the assembly of GUVs in salty solutions by first allowing budding to occur in low salt solutions (Supplementary Fig. 4A). After the formation of buds, salt and then proteins are introduced into the lumens of the surface-attached GUV buds via diffusion (Supplementary Fig. 4B). For these experiments, the assembly and loading steps were done at room temperature.

We first affixed a polydimethylsiloxane (PDMS) gasket (12 mm $\emptyset_{ID} \times 1$ mm height) onto a clean glass slide to form an assembly chamber. We then removed the lipid-coated tracing paper from the vacuum chamber and placed it into the assembly chamber. Next, we added 126 μL of a solution of 119 mM sucrose into the assembly chamber. After three minutes, we added 14 μL of 10× clock buffer underneath the paper. Adding the solution underneath the paper protects the buds that are assembling on the surface of the paper. The concentration gradient equilibrates via diffusion and the components of the clock buffer enter the lumens of the surface-attached buds. Seven minutes later, 10 μL of 15× protein solution was added directly to the external phase, above the paper substrate, and given an additional 110 min for the protein to diffuse into the surface-attached buds (Supplementary Fig. 4C). Supplementary Table 1 summarizes the buffer and protein addition steps for the PAPYRUS-wDL method.

After the incubation, we aspirated the solution in the chamber with a 100–1000 μL pipette set at a volume displacement of 100 μL. We aspirated a total of six times over different locations on the paper. This action detached the buds from the surface of the paper, which self-closed to form GUVs (Supplementary Fig. 4D). The fully-formed GUVs trapped the protein and salts in their interior.

## Sample preparation and imaging

A custom chamber was used for imaging the GUVs. To make this chamber, we broke a PEG-biotin functionalized glass coverslip into two equal parts (~30 mm × 22 mm) by first scoring the surface with a diamond-tipped scriber. We then affixed a circular PDMS gasket ($\emptyset_{OD}$ = 10 mm) with an internal 6 mm × 6 × mm × 1 mm square chamber to one-half of the PEG-biotin functionalized glass coverslip. The PDMS gasket and the glass adhere reversibly through van der Waals interactions if both are clean and free from dust. Next, we added 20 μL of 0.1 mg/mL streptavidin to the chamber and allowed the solution to incubate for 15 min. We then removed and discarded the streptavidin solution and washed the coverslip five times with 60 μL of the sedimentation buffer to remove unbound streptavidin.

We prepared concentration-matched sedimentation buffers by mixing 1260 μL of a solution of 119 mM glucose with 140 μL of the concentrated 10× clock buffer and 100 μL of 15× protein solution in a 1.5 mL Eppendorf tube, to match the concentrations used in the assembly and loading of the GUVs. We added 30 μL of the GUV containing suspension to the chamber and then added 30 μL of the concentration-matched sedimentation buffer. The chamber was sealed with a 22 × 22 mm square coverslip. We waited 3 h for the GUVs to sediment to the bottom of the chamber and bind to the streptavidin-coated glass through biotin-streptavidin interactions.

Prior to imaging, the chamber was flipped upside down, so that the bound GUVs were at the top surface. Flipping allowed us to image using high NA low-working distance objectives on the upright microscope. We used a Zeiss upright microscope (LSM 880, Axio Imager.Z2m, Zeiss, Germany), with a 63× 1.4 NA Oil Plan-Apochromat objective to image the samples. The "red" channel which imaged the Rhod-DOPE lipids was configured with 2.5% power for the 561 nm diode-pumped solid-state (DPSS) laser and a detector gain of 700 A.U. for the confocal photomultiplier tube detector. The "green" channel which imaged the FITC-BSA was configured with 2% power for the 488 nm argon laser and a detector gain of 650 A.U. for the gallium arsenide phosphide (GaAsP) detector. The pinhole diameter was set to 1 A.U., corresponding to a 0.7 μm thick imaging slice. A 7 × 7 tile scan, consisting of 49 images covering a region of 135 × 135 μm per image, was taken using reflection-based autofocusing with an offset of 3 μm into the sample from the glass surface. The image resolution was set to 1584 × 1584 pixels with 4× line averaging. Supplementary Fig. 5 shows a schematic of the imaging geometry.

## Field flatness correction

Although a Plan-Apochromat objective provides good field flatness, there was still noticeable dimming near the edges of the images (Supplementary Fig. 6A). We corrected this dimming by flattening the field. We created a background mask using the red channel and used it on the green channel to obtain the background intensity. We fit this background image with a 2D polynomial, $f(x, y) = a_0 + a_1 x + a_2 y + a_3 x^2 + a_4 y^2 + a_5 xy$. We multiplied the green channel with the flatness correction factor $\left( \frac{z_{max}}{f(x,y)} \right)$. Here $z_{max}$ is the global maximum value of the function $f(x, y)$. An example of the effect of each step of the algorithm on an image is shown in Supplementary Fig. 6B-D.

## Selection using the coefficient of variation (CV) of pixel intensities

GUVs are the target of interest but lipid aggregates, multilamellar vesicles (MLVs), and multivesicular vesicles (MVVs) can also be present in the sample (Supplementary Fig. 7A). We use a selection algorithm that distinguishes GUVs from other objects by their coefficient of variation (CV) of pixel intensities. The CV is the standard deviation of the pixel intensities, $\sigma$ divided by the mean of the pixel intensities, $\mu$, of an object, CV = $\frac{\sigma}{\mu}$. The CV was calculated using the intensity of all pixels located within each object in the red channel. The large contrast between the high pixel values of the bright fluorescent membranes and the low pixel values of the dark lumens results in a high CV value. Aggregates, MLVs, or MVVs have lower CV values than GUVs since the interior of these objects contains membranes that are imaged as bright pixels. Gating for objects with a CV > 0.75 selects mainly GUVs while excluding other objects (Supplementary Fig. 7B). We then examined the images manually and excluded any non-GUV objects that may have been selected. For negative controls where no FITC-BSA was encapsulated in the lumen but was present in the exterior solution, we performed a second automated selection based on the CV of the objects in the green channel to exclude GUVs with invaginations in the membrane.

## Imaging geometry of the lumen

A slice thickness of 0.7 μm with 3 μm offset from the coverslip places the imaging plane within the lumen of GUVs ≥ 4 μm in diameter. Thus, only GUVs with diameters ≥4 μm were considered in our analysis. We measured the intensity of FITC-BSA within a concentric circular region of interest (ROI) at the center of each GUV. The diameter of the ROI is 30% of the GUV diameter, $D_{ROI} = 0.3 D_{GUV}$.

## Determination of empty GUVs

We expect that the hydrodynamic perturbation of adding the buffers with salts or proteins to the chamber will cause some of the buds to detach prematurely and close to form GUVs (Supplementary Fig. 4) Buds that detach from the lipid film and close to form GUVs early in the

process of diffusive loading are expected to remain empty or have very low amounts of encapsulated protein since the GUV membranes are impermeable to the protein. Histograms of the intensities of the lumen of the GUVs show two distinct peaks, one at a lower intensity and the other at an intensity corresponding to the loading concentration (Supplementary Fig. 8A). In comparison, histograms of GUVs that are prepared without any protein show a single low-intensity peak (Supplementary Fig. 8B). We found the location of the peak(s) and the full width at half maximum (FWHM) using the *findpeaks* function for both histograms. The peak position and the right boundary of GUVs not loaded with protein corresponded to the lower peak in the histogram of the samples loaded with protein. The fraction of GUVs within this peak ranged from 18 to 20% of the samples loaded with protein. The fraction did not depend on the loading concentration of the protein [$F(3,8) = 0.46$, $p = 0.72$]. We exclude the bottom 20% of the histogram of all samples prepared with protein as coming from GUVs devoid of protein.

### Bulk measurements of fluorescence intensity of clock reactions

We deposited 50 μL of the reactions into wells of a black clear-bottom 96-well plate. To minimize evaporation during the multiday experiments, we filled empty wells with ultrapure water. We use a SpectraMax® M2e plate reader to measure the mean fluorescence intensity every 30 min for a total of 96 h using the bottom read mode. The chamber temperature was set to 30 °C, fluorescence was excited at 485 nm, and emission was collected between 530 and 538 nm. Measurements were taken with high detector sensitivity and each data point was an average of 6 reads.

### Fluorescence quenching measurements of KaiB-6IAF

To obtain a readout on the state of the PTO using fluorescence measurements, 50 mol% of the KaiB molecules were labeled using 6-(Iodoacetamido)fluorescein (6IAF) to form the fluorescently labeled KaiB-6IAF. KaiB-6IAF is quenched when it forms KaiBC and KaiABC complexes resulting in a drop in the mean fluorescence intensity. When KaiB-6IAF complexes disassociate, they become unquenched, and fluorescence intensity is restored. This allows real-time readout of the state of the clock using measurements of fluorescence intensity. The quenching occurs due to the presence of tryptophan residues near the KaiB-binding site on the CI domain of KaiC[38]. Tryptophan residues quench the fluorescence intensity of many fluorophores, such as the conjugated fluorescein on KaiB-6IAF[39].

We add 25 μL of 50:50 mixture of KaiB:KaiB-6IAF at a concentration of 7 μM to 25 μL of serially diluted KaiC in a black clear bottom 96-well plate. Since KaiB binds slowly to fully phosphorylated KaiC, we obtain kinetic plots of the mean intensity of KaiB-6IAF for 22 h. Note that the formation of the KaiBC complex can occur even without the presence of KaiA because KaiC is initially hyperphosphorylated due to its preparation and storage at low temperatures[27,40,41]. The results in Supplementary Fig. 1A demonstrate the decrease in KaiB-6IAF fluorescence intensity with time for various KaiC concentrations. Supplementary Fig. 1B shows that the decrease in KaiB-6IAF fluorescence intensity is linearly related to KaiC concentration ($y = 0.12[\text{KaiC}] + 1$, $R^2 = 0.97$) (5). The fluorescence intensity decreased by ~60% at 3.5 μM KaiC. Further increasing the concentration of KaiC to 7.0 μM did not result in additional quenching, indicating that the maximal quenching of KaiB-6IAF is ~40 %. This value is consistent with the 42% value reported in the literature for fluorescein-peptide-tryptophan quenching[39]. We thus conclude that at 3.5 μM of KaiC, all available KaiB is bound to KaiC. Indeed, at this concentration, KaiB and KaiC monomer concentrations were equal (3.5 μM KaiB and 3.5 μM KaiC), consistent with reports that KaiBC complexes have a 6:6 monomer ratio[10,42].

### Protein-loading solutions for PTO encapsulation in GUVs

We prepared protein-loading solutions at 15× the intended final protein concentration in 30 μL of 1× clock buffer. The solution was filtered using MilliporeSigma™ Ultrafree™ -MC centrifugal filters in a microcentrifuge at $13,900 \times g$ for 3 min to remove protein aggregates. PTO-GUVs were prepared following the PAPYRUS-wDL protocol.

### Sample preparation for imaging of PTO-GUVs

We followed the sample preparation protocol for GUVs encapsulating FITC-BSA with the following modifications. We did not place a coverslip on the chamber during sedimentation. Instead, we placed the chamber in a lab-built humidity chamber that consisted of two folded Kimwipes saturated with ultrapure water in a 100 mm diameter Petri dish. We find that the use of this ad hoc humidity chamber was sufficient to minimize evaporation of the solution during the 3 h of sedimentation. After three hours, we exchanged the sedimentation buffer with vesicle- and protein-free hydration buffer. Then, we gently removed 30 μL of the supernatant from the sample and added 30 μL of fresh vesicle- and protein-free hydration buffer. This process was repeated five times. Then, the imaging chamber was sealed with a circular glass coverslip (diameter = 12 mm), which produced a 1 mm overhang around the imaging chamber. The overhang was filled with Loctite® Instant Mix Epoxy and allowed to set for at least 15 min before imaging. Sealing with epoxy minimizes evaporation from the chamber over multiple days of imaging.

### Imaging PTO-GUVs

The PTO-GUVs was imaged using dual-channel imaging using a Zeiss LSM 700 upright microscope with a 20× 0.8 NA Plan-Apochromat objective. The sample chamber was flipped so that the immobilized GUVs were close to the objective on the upright stand. We used a Peltier stage to keep the sample at 30 °C. The red channel was configured to image the Rhod-DOPE in the PTO-GUV membranes, and the green channel was configured to image the KaiB-6IAF. The Rhod-DOPE was excited with a 555 nm laser and the KaiB-6IAF was excited with a 488 nm laser. Time-lapse imaging consisted of 10 positions (328 × 328 μm per position) imaged every 2 hours over the course of 100 hours (~4 days). We selected positions that had many PTO-GUVs with polydisperse distributions of diameters. The plane of optimal focus was determined manually at the beginning of the acquisition and a reflection-based autofocus was used to maintain focus over the time-lapse. Images had a resolution of 2048 × 2048 pixels with 4× line averaging. The pinhole was set to the maximum size which is 13.6 Airy Units (AU). The voxel size was 0.16 × 0.16 × 26.7 μm (Supplementary Fig. 9). The pinhole was opened to the maximum since it allowed us to use low laser power to reduce photobleaching over the course of the 4-day experiment.

### PTO time series preprocessing

The raw native *.czi* time-series images were preprocessed using the *MultiStackReg* plugin with the "Translation" algorithm in ImageJ to align the images. The alignment corrected for drift in the images that occur at each acquisition time point. The red channel images, which showed the PTO-GUVs, were segmented from the background using a custom MATLAB routine. Then objects that had mean intensities within ± one full width at half maximum (FWHM) of the global peak in the mean intensity histogram were selected as potential GUVs. We select objects with diameters of 2 ± 0.5, 3 ± 0.5, 4 ± 0.5, 6 ± 0.5, 8 ± 0.5, and 10 ± 0.5 μm for analysis. The selected objects were manually inspected and objects that did not resemble GUVs (defined as spherical objects with uniform intensities) were removed from the analysis. We then used the pixel locations of the PTO-GUVs obtained from the red channel to calculate the mean intensity of KaiB-6IAF within the lumens of the PTO-GUVs in the green channel. Empty GUVs were identified as

GUVs with mean intensities <20% of the encapsulated mean intensity and were not used for analysis.

## Time traces of fluorescence intensity

Time traces of the intensities of KaiB-6IAF from each PTO-GUV were obtained using the *regionprops* MATLAB function. The time trace of the intensity of the background was subtracted from the signal from the PTO-GUVs. The signals were normalized using the intensity at $t^* = 0$. Here $t^* = 0$ represents the time point of the first frame of the time-lapse, which is 7 h after the clock reaction was started by mixing the Kai proteins for loading into the GUVs. The normalized KaiB-6IAF time traces of each PTO-GUV was fit to a two-term exponential decay equation ($y = ae^{bt} + ce^{dt}$). The fitted line was then subtracted from the normalized time traces to correct for photobleaching[43].

## Fast Fourier Transform (FFT) analysis of PTOs

To identify oscillating PTO-GUVs, we performed a fast Fourier transform (FFT) with 1000 point zero-padding on each of the time traces and obtained a single-sided amplitude spectrum. Then, we used the *findpeaks* function to find peaks in the spectrum. We then filtered the spectra to identify spectra with a single global peak that i) had a height > 0.04, ii) was 30 % higher than any other peak, and iii) had a center within a frequency range of $1.39 \times 10^{-5}$ Hz (16 h) to $1.07 \times 10^{-5}$ Hz (30 h). The frequency corresponding to the center of the global peak is converted into the characteristic period of oscillation. Spectra that do not satisfy these criteria are classified as not oscillating. This filtering criteria resulted in ≤2% of the negative control being falsely identified as oscillating. Clock fidelity is the sum of PTO-GUVs that oscillate divided by the total number of PTO-GUVs in the group. A clock fidelity of zero would mean no PTO-GUVs oscillate, and one would mean all PTO-GUVs oscillate.

## Assignment rules for Kai proteins in modeled PTO-GUVs

Kai proteins form complexes during the process of diffusive loading. Approximately 13% of KaiC are expected to be in a KaiABC protein complex (Supplementary Fig. 1A). The KaiABC complex consists of KaiA, KaiB, and KaiC monomers in a 12:6:6 molar ratio[10,42]. KaiA was assigned as dimers, KaiB as tetramers, and KaiC as hexamers. We assigned concentrations in 5000 simulated vesicles by using the *gamrnd* function with the shape parameter $k$ and scale parameter $\theta$. The parameter values were determined assuming a CV of 0.31 and a mean concentration, $\mu$, corresponding to the loading concentration of the protein. Then the parameters were calculated using $k = 1/\text{CV}^2$ and $\theta = \mu \text{CV}^2$. After assignment to each vesicle, the concentration of the constituent components of the KaiABC complexes was redistributed as monomeric KaiA, KaiB, and KaiC.

We assume that KaiB binds to the membrane resulting in a reduction of the free concentration of KaiB in the lumen. We use Eq. (1) with $b = 650$ KaiB monomers per $\mu m^2$ to calculate the concentration of free KaiB ($C_{free, KaiB}$) in a vesicle of radius, $r_i$.

$$C_{i, KaiB_{free}} = C_{i, KaiB}\left(1 - \frac{b}{C_{i, KaiB}N_A}\frac{3}{r_i}\right) \qquad (1)$$

In this equation, $C_{i, KaiB}$ is the nominal concentration of KaiB obtained from the gamma distribution and $N_A$ is Avogadro's number.

## Limiting concentration and ratio rules for modeled PTO-GUVs

Limiting concentrations ($C_{L,[X]}$) and ratios were obtained from our bulk plate reader experiments. The stoichiometric ratios were obtained from values in the literature. The PTO failed in the bulk experiments when the concentration of the PTO proteins was 0.5×, that is when the

concentration of KaiA <0.6 μM, KaiB <1.75 μM, and KaiC <1.75 μM. Following convention in the field, all concentrations are reported as monomeric concentrations. This result was consistent with previous bulk measurements[8,19,22]. The limiting stoichiometric ratios of KaiA and KaiB were measured relative to fixed KaiC concentrations[8,19,22]. The limiting ratio of KaiA to KaiC $0.17 \leq R_{L,[KaiA:C]} \leq 1.02$ and KaiB to KaiC $R_{L,[KaiB:C]} \geq 0.5$. There appears to be no upper limit of stoichiometric ratios for KaiB to KaiC[8,22].

## Calculation of fidelity

A vesicle was considered to oscillate only if (i) all the protein stoichiometries are at or above the limiting ratio and at or below the maximum ratios, and (ii) the concentration of free proteins is at or above the minimum concentration for all protein species. A measure of clock fidelity was determined by taking the sum of vesicles that oscillate divided by the total number of vesicles in the group (5000 simulated vesicles).

## Calculation of periods and amplitude

The period and amplitude of the PTO depend on the concentrations and stoichiometric ratios of the Kai proteins[8]. The concentration of KaiA, KaiB, and KaiC and their ratio varies from vesicle to vesicle. We hypothesized that linear addition to the periods and amplitudes that we measured in our bulk experiment can be used to predict the properties of the PTO in the vesicles. We use Eq. (2), and Eq. (3) and calculate the periods, $T_i$ and amplitude $A_i$ of the encapsulated PTO in vesicle $i$.

$$T_i = T_{[C]} + \Delta T_{[A]:[C]} + \Delta T_{[B]:[C]} \qquad (2)$$

$$A_i = A_{[C]} + \Delta A_{[A]:[C]} + \Delta A_{[B]:[C]} \qquad (3)$$

$T_{[C]}$ and $A_{[C]}$ are the period and amplitude measured from our bulk experiments with concentrations varying from 0.75× to 2.5×, and with the in vitro WT stoichiometry of 0.34:1:1 of monomeric KaiA:KaiB:KaiC. $\Delta T_{[A]:[C]}$, $\Delta A_{[A]:[C]}$ and $\Delta T_{[B]:[C]}$, $\Delta A_{[B]:[C]}$ are the changes in the period and amplitude due to varying KaiA-to-KaiC and KaiB-to-KaiC stoichiometric ratios respectively. $\Delta T_{[A]:[C]}$, $\Delta T_{[B]:[C]}$, and $\Delta A_{[A]:[C]}$ were calculated from the data reported in ref. 8. $\Delta A_{[B]:[C]}$ was obtained from the data reported in Supplementary Fig. 1B. Values between the data points were obtained by linear interpolation. The relationships between the concentration and ratios with the amplitude and period shown in Supplementary Fig. 10.

## PTO in cyanobacteria-mimicking simulations

Cyanobacteria, unlike GUVs, have internal thylakoid membranes[19,44,45]. It is known that approximately 50 % of the KaiB is associated with membrane fractions in cyanobacteria[23]. To simulate the effect of protein variation, we assign the Kai proteins according to the gamma distribution with a mean concentration of KaiA= 2.70 μM, KaiB = 10.80 μM, KaiC = 7.26 μM into 5000 "bacteria" which corresponds to 2.1× the in vitro WT KaiC PTO concentration[4]. The other concentrations were changed accordingly by multiplication. Note the in vivo stoichiometry does not correspond exactly to the in vitro PTO stoichiometry that is commonly used for bulk experiments. We use a CV value of 0.25[17]. Then the KaiB concentration was reduced by 50% before the fidelity, the period, and the amplitude of oscillations were calculated.

For the conditions with SasA and CikA support, we ease the limiting concentration, KaiA <0.3 μM, KaiB <0.9 μM. The limiting ratio of KaiA to KaiC becomes $0.09 \leq R_{L,[A]:[C]} \leq 1.02$ and KaiB to KaiC becomes

$R_{L,[B]:[C]} \geq 0.25$. These numbers were obtained from[8] where CikA and SasA were added to the PTO mixture and ratios and concentration of the Kai proteins were varied.

## Calculation of traces for the "No memory" scenario

We select from the gamma distribution 17 times to reflect 17 cycles. We calculate the period, $T_i$, and expected amplitude, $A_i$, for vesicle $i$ for each of the $j = 1 - 17$ cycles. We then obtain the time trace for each cycle using $F_{i,j} = A_i \sin\left(\frac{2\pi}{T_i} t\right)$, and concatenate the trace for each cycle to obtain the time trace for the simulated cyanobacteria. $t = 0 : 0.01 : T_i$. $t$ and $T_i$ are measured in hours. The simulation was repeated 5 times, and the mean and standard deviation of the parameters were used to plot the data in Fig. 7.

## Calculation of traces for the "Perfect memory" scenario

We retain the initial concentration obtained from the gamma distribution for each of the 17 cycles. We calculate the period, $T_i$, and expected amplitude, $A_i$, for vesicle $i$ for each of the $j = 1 - 17$ cycles. We then obtain the time trace for each cycle using $F_{i,j} = A_i \sin\left(\frac{2\pi}{T_i} t\right)$, and concatenate the trace for each cycle to obtain the time trace for the simulated cyanobacteria. $t = 0 : 0.01 : T_i$. $t$ and $T_i$ are measured in hours. The simulation was repeated 5 times, and the mean and standard deviation of the parameters were used to plot the data in Fig. 7.

## Calculation of traces for the "No memory + TTFL" scenario

To simulate the effect of the TTFL, we assume that the stoichiometry is corrected so that at the end of each cycle, the simulated cyanobacteria will have a period and amplitude equal to the WT average. We averaged the period and amplitude calculated from the concentration and stoichiometry of the protein at the beginning of each cycle with the WT period and used the averaged value for $A_i$ and $T_i$.

## Calculation of phase shifts

The time traces of each individual vesicle are averaged. Then a sinusoid, $f(t) = B \sin\left(\frac{2\pi}{t_0} t - \phi\right)$ was fit to the curve to obtain the characteristic period, $t_0$. In this equation, $B$ is the amplitude and $\phi$ is the phase shift. The use of a sine simply shifts the phases by $\frac{\pi}{2}$ relative to the cosine which was used in refs. 3,8,11. Then, each signal is windowed to a 24-hour cycle. The windowed signal for each simulated cyanobacteria is fitted with the sine function with $t_0$ as a fixed parameter to obtain the phase shift, $\phi$, and the amplitude, $B$. For the polar plots in Fig. 7, the reference period and amplitude are calculated using the mean clock protein concentration and stoichiometric ratios. We randomly chose 500 simulated cyanobacteria from the 5000 to plot the polar plots.

## Calculation of the synchronization index

To characterize the synchronization of the signals, we used the synchronization index as described in refs. 11,31. The synchronization index (SI) is based on the Shannon entropy of the phase distribution, defined as $SI = (S_{max} - S) / S_{max}$. Here $S_{max} = \ln N$, where $N$ is the number of bins and $p_k$ is the probability of the data occurrence in each bin. The entropy of the distribution is defined as $S = -\sum_{k=1}^{N} p_k \ln p_k$. Following[11], we used $N = 8$ bins for the phase distributions as a function of time. The phase shifts were wrapped to be between 0 and $2\pi$. The SI was calculated for the signal for each 24-hour window. The SI is set up where a value of 1 corresponds to perfect synchronization (Dirac-like distribution) and a value of 0 corresponds to no synchronization (uniform distribution)[31].

## Statistics and reproducibility

Sample sizes and the number of repeats are reported in the legends and the main text. Statistical tests were performed using MATLAB 2023b (version 23.2). A $p$ level of >0.05 was considered significant.

## Reporting summary

Further information on research design is available in the Nature Portfolio Reporting Summary linked to this article.

## Data availability

All data supporting the results and conclusions are available within this paper and the Supplementary Information. Source data are provided with this paper.

## Code availability

The custom codes that support the findings within this paper and other findings of this study are available on Zenodo at (doi: 10.5281/zenodo.15455022), (doi: 10.5281/zenodo.15455000), and (doi: 10.5281/zenodo.15455022). All codes are released under the MIT license, allowing for free use, modification, and distribution.

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

## Acknowledgements

This work was partially supported by the National Science Foundation through NSF CAREER DMR-1848573 (to A.B.S), US National Institutes of Health (NIH) grants R35GM144110 (to A.L.), and US Army grant W911NF-23-1-0248 (to A.L.). Support was also provided by NSF-CREST: Center for Cellular and Biomolecular Machines at the University of California, Merced (NSF-HRD-1547848) (to A.Z.T.L). The data in this work was collected, in part, with a confocal microscope acquired through the National Science Foundation MRI Award Number DMR-1625733 (to A.B.S), that is housed and managed by the Imaging and Microscopy Core at UC Merced. We thank Joseph Pazzi, Joel Heisler, Archana Chavan, and Supratim Dey for technical assistance and for valuable discussions.

## Author contributions

A.B.S. and A.L conceptualized the study. A.Z.T.L performed all experiments and analysis. A.B.S formulated the model. A.Z.T.L implemented the model and obtained results. A.Z.T.L wrote the first draft. All authors contributed to the final draft.

## Competing interests

The authors declare no competing interests.
