## [Transparent Peer Review file · Nature Communications]

Reconstitution of circadian clock in synthetic cells reveals principles of timekeeping

Corresponding Author: Professor Anand Subramaniam

Version 0:

Reviewer comments:

Reviewer #1

(Remarks to the Author)

The post-translational cyanobacterial circadian oscillator had previously generated major interest as it showed robust circadian rhythms based on protein phosphorylation/interactions. In particular the oscillator was shown to also work in vitro when mixing the proteins KaiA, KaiB, KaiC.

In the present work, the authors study the behavior of the oscillator when encapsulated in vesicles. The study appears to be the first such study of compartmentalization of this oscillator and will be of interest to researchers in the field.

Specific suggestions:

- the authors might consider to show more about the experimental setup, how the Kai oscillator works, how it is encapsulated and observed in thousands of vesicles in the first Figure (now in the SI) to introduce readers who are not so familiar with it into the topic. Only showing the characterization of the protein distribution (now Fig. 1) is quite a technical start of a paper that is about oscillators
- the term "clock fidelity" might be a little misleading as you appear to look at the fraction of oscillating GUVs. The term let's one expect a statement about how well the time is kept/variations in clock period.
- can you indicate which size range of the GUVs corresponds to *S. elongatus*?
- when you discuss KaiABC binding to the membrane - is the membrane composition comparable between GUVs and *S. elongatus* (or does it matter)?
- does the oscillator run slow over time due to exhaustion of fuel (ATP?) – and doesn't this have a major effect on de-synchronization?
- it is not clear what the authors actually modeled for Fig. 5 and 6 – is there a set of ODEs that you solved – where is it?
- as the finding of the stabilizing effect is only found in the simulations (and is not part of the experiments), the authors should consider to just briefly discuss that point and refer to the SI. In particular, the TTFL is not present in the experiments, and the model simply assumes a correction of stoichiometry, but does not explicitly model the underlying reactions.
- why should cyanobacteria bother about synchrony in the population? Aren't there external triggers (light) that reset the clock.

(Remarks on code availability)

Reviewer #2

(Remarks to the Author)

Li and co-authors present an interesting experimental study that encapsulates the circadian proteins from cyanobacteria, KaiA, KaiB and KaiC, in GUVs to examine the impact of confinement and reduced volume on the oscillatory behavior of the proteins. This work is a nice bridge between the many studies that examine the oscillatory PTO behavior in bulk and those that focus on in vivo studies. However, there are several technical issues that need to be addressed to properly validate the work to be impactful.

1. Using BSA to validate PAPYRUS-wDL is necessary but not sufficient to characterize the encapsulation properties of the Kai proteins. What evidence do the authors have to support that their results presented in Fig 1 apply to the Kai proteins? Especially since the authors demonstrate that BSA does not localize to the GUV membranes but Kai proteins do. This finding suggests that the encapsulation efficiency and distributions may be different between BSA and Kai. the authors need to repeat experiments with Kai proteins or provide clear rationale and supporting evidence that the results are expected to be the same.

2. Following on #1, the authors claim that the concentrations are gamma distributed with variations that are independent of concentration. Both of these statements are not well supported by the data shown in Fig 1A. the 2 higher concentrations do not appear to be described by gamma distributions. The 2.63 uM data appears bimodal and the 4.5 uM has amplified skewness and spread in the data. This spread is also seen in Figs 1B,C that show that the highest concentration data displays much larger spread in the data than lower concentrations. How do the authors explain this variation? and how is it taken into account in experiments? is the mean a valid parameter?

3. The authors make strong claims that their results can explain in vivo results and mechanisms underlying oscillation period, robust and synchrony. However, cyanobacteria are ~2 um and most of the GUVs are larger than this size. The 3 example GUVs in Fig 2 are all ~10 um or higher. Why not choose examples that are closer in size to cyanobacteria?

4. The membrane association is not well supported or described. Do membrane-bound KaiBs participate in KaiABC oscillations? The authors seem to indicate that they do not contribute, but how do they draw that conclusion? Or if they do participate then the argument that the membrane-bound Kais reduce the effective concentration needs to be revised. How does the association depend on GUV membrane properties? Can authors block the interaction or introduce specific interactions to determine the specificity of the association? A more careful investigation of this effect and the implications on their results is needed.

5. To understand the size effects the authors should show data in Fig 2B for other sizes in SI.

6. Why is the average period not 24 hrs but is closer to 22 hrs?

(Remarks on code availability)

Reviewer #3

(Remarks to the Author)

The manuscript by Li et al represents an important study of the fundamental biophysical properties of the cyanobacterial circadian oscillator proteins encapsulated at the size scale of a bacterial cell. Though previous studies have attempted to address these issues in living cells, this work has the virtue of using a rigorously defined in vitro system to analyze the effect of encapsulation. An important effect that the authors find is that the surface area/volume ratio is a key predictor of oscillator function and is associated with KaiB interaction with the membrane. Overall, my judgment is that the manuscript is of high quality and presents important results for the field. I have some suggestions for additional experiments and analysis that could strengthen the conclusions:

1. The authors show quite convincingly that the fraction of oscillating vesicles increases dramatically with increasing Kai protein concentration from 1x to 2.5x, and they interpret this to be caused in part by KaiB interaction with the membrane. Does the fraction of oscillating vesicles ("fidelity") also increase if only the concentration of [KaiB] is increased?

2. Related: is there any correlation between the intensity of KaiB fluorescence on the membrane and whether a vesicle (of a given size) oscillates or not? and is it possible to detect any oscillation of the membrane fluorescence or does this really represent a "stuck" pool of KaiB?

3. I don't consider this essential, but worth considering: I find the model-based conclusions about the role of SasA and CikA suggestive. It should be possible to test this directly since it has been recently shown that SasA and/or CikA can be incorporated successfully into the in vitro oscillator (Chavan, 2021)

minor comment:

the authors write "The mean amplitudes of the oscillations, which correlate with the fraction of KaiC that participates in the clock reaction..." is there a citation that shows this clearly? I could imagine other explanations e.g. reduced synchrony between KaiC hexamers all of which are still participating in the reaction. I think if there is not clear evidence that shows this correlation, the sentence should be edited.

(Remarks on code availability)

Version 1:

Reviewer comments:

Reviewer #1

(Remarks to the Author)

The authors have addressed all of the reviewers' comments appropriately. However, this reviewer still is slightly confused about the mathematical model for the oscillations. Perhaps the authors could clarify their model even further. Specifically, is it correct that they essentially draw a number probabilistically from a gamma distribution estimated from their experimental results and then simply classify the droplets as oscillating or non-oscillating based on protein concentrations, again determined from their experimental results? In other words, there is no physical or mechanistic model of the oscillator in this case. If so, isn't it somewhat surprising that stoichiometry does not seem to play a major role? Or does it? This should particularly influence multimerization reactions that are involved in the PTO.

(Remarks on code availability)

Reviewer #2

(Remarks to the Author)

The authors have adequately addressed my comments. I recommend publication

(Remarks on code availability)

Reviewer #3

(Remarks to the Author)

I thank the authors for their thoughtful consideration of my previous comments. I do think it would add something substantial to the paper to test experimentally the prediction that increasing [KaiB] only increase "fidelity" rather than just in a model. But if this is prohibitive due to personnel constraints (implied in the response letter), my judgment is that this is a valuable paper that is worth publishing already in its current form.

(Remarks on code availability)

Version 2:

Reviewer comments:

Reviewer #1

(Remarks to the Author)

The authors appropriately replied to my final remarks and the manuscript may now be accepted for publication.

(Remarks on code availability)

Reviewer #1 (Remarks to the Author):

Reviewer: *The post-translational cyanobacterial circadian oscillator had previously generated major interest as it showed robust circadian rhythms based on protein phosphorylation/interactions. In particular the oscillator was shown to also work in vitro when mixing the proteins KaiA, KaiB, KaiC.*

In the present work, the authors study the behavior of the oscillator when encapsulated in vesicles. The study appears to be the first such study of compartmentalization of this oscillator and will be of interest to researchers in the field.

Response: We thank the reviewer for their careful reading of our manuscript and for their kind words and constructive comments.

Specific suggestions:

Reviewer:- *the authors might consider to show more about the experimental setup, how the Kai oscillator works, how it is encapsulated and observed in thousands of vesicles in the first Figure (now in the SI) to introduce readers who are not so familiar with it into the topic. Only showing the characterization of the protein distribution (now Fig. 1) is quite a technical start of a paper that is about oscillators*

Response: We have introduced a new Fig. 1 that shows the experimental flow and experimental setup from the process of assembly of PTO-GUVs to imaging and data analysis. The new figure should help the reader who is unfamiliar with this topic. We thank the reviewer for this helpful suggestion that has made the paper stronger.

Reviewer:- *the term “clock fidelity” might be a little misleading as you appear to look at the fraction of oscillating GUVs. The term let’s one expect a statement about how well the time is kept/variations in clock period.*

Response: We thank the reviewer for raising this point about other potential definitions of fidelity. In our revised manuscript, we have added quotes around the phrase clock fidelity to clarify our specific definition for the term as applied to the population of circadian clocks in PTO-GUVs. We also moved the definition of this term to our Introduction for additional clarity.

“To capture this behavior, we define a “clock fidelity” parameter. The fidelity is the number of oscillating PTO-GUVs divided by the total number of PTO-GUVs. A fidelity of zero would mean no PTO-GUVs oscillated and a fidelity of one would mean all of them oscillated.”

Reviewer:- *can you indicate which size range of the GUVs corresponds to *S. elongatus*?*

Response: We have added a statement that the volume of two-micron diameter GUVs

corresponds to the volume of *S. elongatus* (in the Results section under the “Reconstitution of PTOs in GUVs” subsection). We thank the reviewer for bringing this unintentional omission to our attention.

Reviewer:- *when you discuss KaiABC binding to the membrane - is the membrane composition comparable between GUVs and S. elongatus (or does it matter)?*

Response: We thank the reviewer for this question. The composition of the membrane of the GUVs and cyanobacteria are different. The membranes of cyanobacteria consist of monogalactosyldiacylglycerol (MGDG), digalactosyldiacylglycerol (DGDG), sulfoquinovosyldiacylglycerol (SQDG), and phosphatidylglycerol (PG). MGDG and DGDG compose >80 mol% of lipids in the plastid envelope and >90 mol% of lipids in the thylakoid membranes. The membranes of the GUVs are composed primarily of dioleoyl-*sn*-glycero-phosphatidylcholine (DOPC). DOPC is used widely in *in vitro* reconstitution experiments in the synthetic cell community, including by us, due to the ease that these lipids form cell-sized GUVs. We observe experimentally that KaiB binds to DOPC membranes similar to the reports of KaiB binding to membranes of cyanobacteria in Kitayama et al. (23). We surmise that the lipid composition that we use in our GUVs can recapitulate KaiB-membrane binding despite the different composition compared to the lipid membranes of cyanobacteria.

Reviewer:- *does the oscillator run slow over time due to exhaustion of fuel (ATP?) – and doesn't this have a major effect on de-synchronization?*

Response: We thank the reviewer for raising this question. Fig. S6 of Rust et al. (doi: 10.1126/science.1197243) reports that the period of the *in vitro* KaiABC oscillator increases slightly (by approximately 2 hours) as the ATP % decreases from 100 % to 50 %. Chavan et al. (8) found that the concentration ATP dropped by approximately 25% over 115 hours for conditions that we report here. The duration of their experiments was approximately 15 hours longer than our experiments. Thus, we do not expect the ATP to be exhausted and expect only a slight lengthening of the period from the initial period (to a maximum of one hour) over the course of 100 hours (0.01 hr per hour of experiment). This relatively small change in period over the course of the experiment is not expected to have a major effect on de-synchronization. The de-synchronization that we observe is due to the variation in periods between the PTO-GUVs with slightly different stoichiometries of the Kai proteins during the process of encapsulation.

Reviewer:

1) *it is not clear what the authors actually modeled for Fig. 5 and 6 – is there a set of ODEs that you solved – where is it?*

Response: We thank the reviewer for this question. We do not solve a coupled set of ODEs. We developed a novel phenomenological model in this paper that determines via conditional rules if

a particular composition of Kai proteins will oscillate, and the period and amplitude of the oscillation. The conditional rules are obtained from bulk *in vitro* experiments that we and others have reported. We then apply these rules to the simulated distribution of proteins in individual cell-sized PTO-GUVs to determine if the PTO-GUVs will oscillate and the period and amplitude of the oscillation. The phenomenological model does not require the use of assumptions of the molecular details of the half reactions that compose the clock reaction.

Our phenomenological model additionally allows us to model the effects of time-dependent changes in the period and amplitude by using the expected changes in concentration of Kai proteins at each cell division (now Figure 7). Although in principle it is possible to input the simulated protein concentrations into various coupled ODE models that have been used to model the oscillator, we will not obtain any new insights into the expected effect of variation in concentration and the effect of confinement on the behavior of oscillators individually or in a population. Use of experimentally measured behavior of the clock protein concentration on the oscillation characteristics avoids still not agreed upon assumptions and simplifications of the molecular details that are used in ODE models.

2) - *as the finding of the stabilizing effect is only found in the simulations (and is not part of the experiments), the authors should consider to just briefly discuss that point and refer to the SI. In particular, the TTFL is not present in the experiments, and the model simply assumes a correction of stoichiometry, but does not explicitly model the underlying reactions.*

Response: We thank the reviewer for this comment. We agree with the reviewer that we do not reconstitute the TTFL in our experiments. It will be extremely difficult to test a TTFL in an experimental platform with current technology. Controlled transcription and translation in *in vitro* cell-free systems is still in relative infancy.

Given the challenges of reconstituting an experimental model of TTFL control of the PTO, we believe the section on the stabilizing effect of the TTFL on the PTO shown by the phenomenological model is important to be included in the main text instead of the SI since it provides important insights. The model puts limits on how cyanobacteria maintain the “fidelity” and synchrony of the clock *in vivo*. It bridges current *in vitro* understanding with *in vivo* results.

It has been known since Kondo first reported successful reconstitution of the Kai oscillator in *in vitro* bulk experiments that the oscillations are extremely robust and stable even at sub *in vivo* concentrations (0.75x), opening the question of why do cells express Kai proteins at much higher levels *in vivo* than necessary to sustain oscillations *in vitro*. Our work shows that high levels are necessary to overcome partitioning errors in cell-like volumes and losses due to sequestration of a fraction of KaiB to membranes. The section on the TTFL shows that even when the partitioning errors that cause the oscillator to stop oscillating are overcome with higher concentrations, differences in protein concentrations will still lead to desynchronization between the cells. Our work shows that the TTFL is obligatory to maintain synchrony between cells but not necessary for all cells to oscillate. These results, obtained using our phenomenological model, agrees with the *in vivo* results of Teng, S.W. et al (11). It was not our goal to model the molecular

mechanisms of the TTFL, rather to observe the effect of correcting stoichiometry by the TTFL on the synchronization of the PTO. Given the still incomplete quantitative understanding of the TTFL from a molecular perspective, we believe our implementation of the TTFL captures qualitatively what is believed to occur in cyanobacteria.

Reviewer:- *why should cyanobacteria bother about synchrony in the population? Aren't there external triggers (light) that reset the clock.*

Response: We thank the reviewer for posing this interesting question. This reviewer is correct that environmental cues, especially light, entrain the clocks of individual cyanobacteria such that the population is synchronized. Nevertheless, even in the absence of external cues and intracellular communication, cyanobacteria can maintain the synchrony of their internal clock (3). Our work provides an experimentally supported platform and phenomenological model that explains how cyanobacteria are able to achieve this fascinating feat.

Reviewer #2 (Remarks to the Author):

Reviewer: *Li and co-authors present an interesting experimental study that encapsulates the circadian proteins from cyanobacteria, KaiA, KaiB and KaiC, in GUVs to examine the impact of confinement and reduced volume on the oscillatory behavior of the proteins. This work is a nice bridge between the many studies that examine the oscillatory PTO behavior in bulk and those that focus on in vivo studies. However, there are several technical issues that need to be addressed to properly validate the work to be impactful.*

Response: We thank this reviewer for his/her constructive comments on our manuscript.

Reviewer: *1. Using BSA to validate PAPYRUS-wDL is necessary but not sufficient to characterize the encapsulation properties of the Kai proteins. What evidence do the authors have to support that their results presented in Fig 1 apply to the Kai proteins? Especially since the authors demonstrate that BSA does not localize to the GUV membranes but Kai proteins do. This finding suggests that the encapsulation efficiency and distributions may be different between BSA and Kai. the authors need to repeat experiments with Kai proteins or provide clear rationale and supporting evidence that the results are expected to be the same.*

Response: We appreciate the reviewer's feedback. While this is a good suggestion, there are several reasons that complicate the direct measurement of encapsulation of the Kai clock proteins experimentally, which led us to use BSA as our model protein to measure encapsulation statistics.

As we show in our experiments, the fluorescence intensity of KaiB-6IAF in an individual GUV is a function of the concentration of the Kai proteins and their location in the temporal sequence of the oscillation. Thus, the fluorescence intensity of KaiB-6IAF is not only a function of the concentration. The requirement of being able to correlate intensity uniquely to concentration prevents us from determining the distribution of Kai proteins directly in a population of GUVs. We thus chose the widely used soluble protein BSA, which does not participate in complexation reactions or binding to the membrane at the concentrations that we use, as our model protein to determine partitioning statistics.

Further, if we assume that KaiB-6IAF follows the partitioning statistics of BSA but does not bind to the membrane, we lose the dependence of the fidelity on the size of the GUVs that we observe in our experiments, while still retaining the dependence on concentration (compare **Fig. R1** below with Fig. 4C). Only by incorporating membrane binding can we reproduce the experimental observations. The convincing demonstration of the collapse of the fidelity data when plotted against the SA/V ratio showing a surface area effect (now Fig. 4D) and the recapitulation of experimental observations of size dependence shows that our assumption — that Kai follows the encapsulation statistics of BSA but then a fraction attaches to the membrane — is reasonable.

Figure R1. Modeled clock fidelity in PTO-GUVs with no KaiB membrane binding. KaiB binding coefficient is set to zero. All GUV sizes have the same fidelity in this scenario.

Reviewer: 2(a). Following on #1, the authors claim that the concentrations are gamma distributed with variations that are independent of concentration. Both of these statements are not well supported by the data shown in Fig 1A. the 2 higher concentrations do not appear to be described by gamma distributions. The 2.63 μM data appears bimodal and the 4.5 μM has amplified skewness and spread in the data. This spread is also seen in Figs 1B,C that show that the highest concentration data displays much larger spread in the data than lower concentrations. How do the authors explain this variation? and how is it taken into account in experiments? is the mean a valid parameter?

Response: We thank the reviewer for these comments.

We had considered alternative models that could describe the distribution of protein concentration in GUVs. We reproduce two goodness of fit (GOF) parameters, the adjusted R^2 and the root mean square error (RMSE) values for the fit of the datasets with a gamma distribution, a one-term Gaussian distribution, and a two-term Gaussian distribution (Table R1). Fits were performed using the Curve Fitter app in MATLAB. Excepting the data set obtained for a loading concentration of 2.63 μM , the RMSE values are lower for the gamma distribution and the adjusted R^2 values are higher, indicating a better fit to the data. The two term Gaussian, which could be used to describe bimodal distributions, has a lower RMSE and higher adjusted R^2 values for some concentrations. However, the need to incorporate additional fitting parameters and the modest change in the GOF suggest using two terms is overfitting the data and is not illuminating the underlying distribution that describes the experimental data.

We agree with the reviewer that the data for the 2.63 μM loading concentration appears to have two peaks. All four distributions were obtained by averaging the distributions of $N=3$ independent repeats of the experiments to test for reproducibility and to account for potential

sample-to-sample variations in protein addition, diffusion conditions, and imaging conditions that can occur even with careful experimental control. Only the distributions prepared with 2.63 μM of protein appears to show two peaks. Despite the apparent two peaks, the gamma distribution fit of the 2.63 μM data, which were performed identically to the other fits, reproduces the trends we observe in the coefficient of variation (CV) and linear correspondence of the mean encapsulated concentration with the loading concentration, across all four loading concentrations (now Fig. 2B, C).

Concentration (μM)	Gamma		One Term Gaussian		Two Term Gaussian	
	Adjusted R^2	RMSE	Adjusted R^2	RMSE	Adjusted R^2	RMSE
0.88	0.9962	0.0151	0.9805	0.0343	0.9904	0.240
1.75	0.9772	0.0233	0.9445	0.0364	0.9837	0.0197
2.63	0.8976	0.0455	0.9337	0.0366	0.9452	0.033
4.50	0.98299	0.016495	0.97494	0.02016	0.9875	0.0140

Table R1. Goodness of fit parameters to the data.

We additionally show in Table R2 a comparison between the mean concentration and CV obtained from the gamma distribution fit and the nonparametric mean and coefficient of variation directly from the experimental data. The values obtained by not assuming any specific distribution is comparable to the values obtained from the fitted parameters of the gamma distribution.

Concentration (μM)	Gamma Fit		Nonparametric	
	Mean	CV	Mean	CV
0.88	0.9	0.35	0.9	0.34
1.75	1.9	0.38	1.8	0.36
2.63	2.6	0.33	2.7	0.29
4.50	4.6	0.25	4.5	0.26

Table R2. Comparison between the mean and coefficient values obtained from the parameters obtained by the fitted gamma distribution and those obtained directly from the data without assuming a specific distribution.

Regarding the “amplified skewness” and spread for the 4.5 μM dataset, the appearance of amplified skewness is expected due to the much higher magnitude in the concentrations, nearly two times higher than 2.63 μM . The CV, which is a measure of variation (spread), normalizes for the expected increase in the absolute magnitude of the variance with the mean magnitude of the values of the histogram. Fig. 2B shows that the CV values for the 4.5 μM dataset is similar to the other concentrations indicating that there is no unexpected ‘amplified skewness’. Additionally, the RMSE values and the adjusted R^2 values in Table R1 show that the gamma distribution describes this data well.

Regarding the spread in the data in the now Figure 2C, calculating the CV values of the three data points for each concentration in Figure 2c shows that for the 0.88 μM dataset the CV is 0.13, for the 1.75 μM dataset the CV is 0.11, for the 2.63 μM dataset the CV is 0.06, and for the 4.5 μM dataset the CV is 0.14. These values show that there is not much larger spread in the data and the results are comparable within expected experimental variation.

Reviewer: 3. *The authors make strong claims that their results can explain in vivo results and mechanisms underlying oscillation period, robust and synchrony. However, cyanobacteria are ~2 μm and most of the GUVs are larger than this size. The 3 example GUVs in Fig 2 are all ~10 μm or higher. Why not choose examples that are closer in size to cyanobacteria?*

Response: We thank the reviewer for this comment. The primary purpose of Fig. 2 (now Fig. 3) was to choose visually clear images of our experiments showing the GUVs and the oscillation in fluorescence intensity. These happen to be easiest to demonstrate in the largest GUVs. In our revised manuscript, we have added examples with the images and signals of GUVs of sizes 2-3 μm , 4-6 μm , 8-10 μm in Supplementary Figure 2 to address the reviewer's comment.

Reviewer: 4. *The membrane association is not well supported or described. Do membrane-bound KaiBs participate in KaiABC oscillations? The authors seem to indicate that they do not contribute, but how do they draw that conclusion? Or if they do participate then the argument that the membrane-bound Kais reduce the effective concentration needs to be revised. How does the association depend on GUV membrane properties? Can authors block the interaction or introduce specific interactions to determine the specificity of the association? A more careful investigation of this effect and the implications on their results is needed.*

Response: Our assumption is that membrane-bound KaiB does not participate in the reaction. This assumption is supported by the apparent negative role that the membrane plays in the fidelity of the PTO (see **Fig. 4D**). We have made an addition to the revised manuscript to explicitly state this assumption. KaiB has two different protein-protein interfaces that it uses simultaneously to bind to KaiC and KaiA. When bound to the membrane, it is likely that at least one of these interfaces is sterically hindered. It is thus hard to imagine that membrane-bound KaiB can form the necessary complexes to participate in the clock reaction. With the assumption that membrane-bound KaiB does not participate in the clock reaction, we are able to recapitulate experimental data. How the membrane properties affect the association of KaiB with the membrane is currently unknown. Indeed, prior to this work, the effect of membrane binding of KaiB on the function of the clock was not considered. We are hoping that the publication of this work, which shows the importance of the membrane on the behavior of the clock, will spur the careful investigations as suggested by the reviewer into this hitherto understudied aspect of Kai protein biochemistry.

Reviewer: 5. *To understand the size effects the authors should show data in Fig 2B for other sizes in SI.*

Response: We thank the reviewer for this suggestion and have provided the images and clock signals for other sizes of vesicles in Supplementary Figure 2.

Reviewer: 6. *Why is the average period not 24 hrs but is closer to 22 hrs?*

Response: We appreciate the reviewer's question. The natural period of the *in vitro* PTO is not 24 hours. Kageyama et al. (19) reports the period as being 22 hours and Chavan et al. (8) reports that depending on stoichiometry of the proteins, the period ranges from 20 to 27 hours. Thus, the average period that we report of 22 to 23 hours is consistent with the known behavior of the PTO oscillator.

Reviewer #3 (Remarks to the Author):

Reviewer: *The manuscript by Li et al represents an important study of the fundamental biophysical properties of the cyanobacterial circadian oscillator proteins encapsulated at the size scale of a bacterial cell. Though previous studies have attempted to address these issues in living cells, this work has the virtue of using a rigorously defined in vitro system to analyze the effect of encapsulation. An important effect that the authors find is that the surface area/volume ratio is a key predictor of oscillator function and is associated with KaiB interaction with the membrane. Overall, my judgment is that the manuscript is of high quality and presents important results for the field. I have some suggestions for additional experiments and analysis that could strengthen the conclusions:*

Response: Thank you for your helpful questions and comments.

Reviewer: *1. The authors show quite convincingly that the fraction of oscillating vesicles increases dramatically with increasing Kai protein concentration from 1x to 2.5x, and they interpret this to be caused in part by KaiB interaction with the membrane. Does the fraction of oscillating vesicles ("fidelity") also increase if only the concentration of [KaiB] is increased?*

Response: We thank the reviewer for this question. Yes, we would expect the fidelity to increase with increasing KaiB concentration. Our model shows that when the concentration of KaiB is increased by 2 \times and 10 \times of the cell-like concentration (8.75 μ M) to 17.5 μ M and 87.5 μ M, the fidelity of the cell-like GUVs (2 μ m) increased from 70 % to 96.8 % and 97.8 % respectively (**Fig. R2**). *In vivo* the concentration of KaiB is reported to be approximately 9.0 μ M by Chew et al. (4), compared to the 8.75 μ M KaiB for our cell-like (2.5 \times) concentration. Thus, the needed KaiB concentrations to obtain very high fidelities are significantly above what is reported *in vivo*.

Figure R2. Clock fidelity model when KaiB concentration is increased. Clock fidelity when KaiB concentration is set to 8.75 μM (blue), 17.5 μM (orange), and 87.5 μM (red), as a function of vesicle diameter. KaiA and KaiC are kept at 3.0 μM and 8.75 μM (2.5 \times), respectively.

Reviewer: 2. *Related: is there any correlation between the intensity of KaiB fluorescence on the membrane and whether a vesicle (of a given size) oscillates or not? and is it possible to detect any oscillation of the membrane fluorescence or does this really represent a "stuck" pool of KaiB?*

Response: We thank the reviewer for this intriguing question. Indeed, it would be interesting to know if KaiB oscillates to and from the membrane as is suggested to occur *in vivo*. We were not able to isolate the intensity of KaiB-6IAF on the membrane in our time lapse experiments. We use a 20x objective with a low numerical aperture of 0.8, low pixel dwell time, low laser power, and open the pinhole to allow maximal light into the detector to minimize the photobleaching of KaiB-6IAF while obtaining sufficient signal to reconstruct the oscillations over 4 days. This experimental setup limits the signal-noise ratio and prevents us from isolating the pixels that correspond to the membrane from the pixels in the lumen. Our static images in Fig. 5 were taken with a 63x objective with a numerical aperture of 1.4 which allows us to isolate the pixels that correspond to the membrane from the lumen.

Reviewer: 3. *I don't consider this essential, but worth considering: I find the model-based conclusions about the role of SasA and CikA suggestive. It should be possible to test this directly since it has been recently shown that SasA and/or CikA can be incorporated successfully into the in vitro oscillator (Chavan, 2021)*

Response: We thank the reviewer for this question. While we agree it would be nice to perform the experiment with SasA and CikA, Li has graduated and is no longer at UC Merced and the difficulty of setting up the time-lapse experiments prevents us from easily testing the role of SasA and CikA experimentally. We are aiming to encapsulate the full clock with SasA, CikA, RpaA and DNA to study in more detail the intricacies of the full-clock on controlling gene expression in cell-size compartments in future work.

Reviewer: *minor comment:*

the authors write "The mean amplitudes of the oscillations, which correlate with the fraction of KaiC that participates in the clock reaction..." is there a citation that shows this clearly? I could imagine other explanations e.g. reduced synchrony between KaiC hexamers all of which are still participating in the reaction. I think if there is not clear evidence that shows this correlation, the sentence should be edited.

Response: We thank the reviewer for this comment. Nakajima et al. (7) shows that the amplitude of the KaiABC oscillation is directly related to the amount of KaiC that participates (i.e.,

phosphorylates). Our study measures fluorescence from labeled KaiB instead of KaiC phosphorylation. This fluorescence is quenched upon KaiB-KaiC binding, which can only occur when KaiC is functional and participating (see Heisler et al. (14)). If no KaiC participates in a reaction, then the amplitude of the oscillation is zero. However, the reviewer is also correct that if a PTO reaction is 100% desynchronized the amplitude of the oscillation will also be zero. In our revised manuscript we cite reference (7) after the sentence in question.

Reviewer #1 (Remarks to the Author):

Reviewer: *The authors have addressed all of the reviewers' comments appropriately. However, this reviewer still is slightly confused about the mathematical model for the oscillations. Perhaps the authors could clarify their model even further. Specifically, is it correct that they essentially draw a number probabilistically from a gamma distribution estimated from their experimental results and then simply classify the droplets as oscillating or non-oscillating based on protein concentrations, again determined from their experimental results? In other words, there is no physical or mechanistic model of the oscillator in this case. If so, isn't it somewhat surprising that stoichiometry does not seem to play a major role? Or does it? This should particularly influence multimerization reactions that are involved in the PTO.*

Response: Yes, we draw a number probabilistically from a gamma distribution. We classify droplets as oscillating and non-oscillating based on protein concentration and protein stoichiometry. We incorporate stoichiometry in our model since experimental data shows that if the ratio of KaiA to KaiC is not within $0.17 \leq R_{L,[KaiA:C]} \leq 1.02$ and KaiB to KaiC is not within $R_{L,[KaiB:C]} \geq 0.5$ the PTO will not oscillate. There appears to be no upper limit of stoichiometric ratios for KaiB to KaiC^{8,22}. Thus, the reviewer is correct that stoichiometry plays a major role as has been shown experimentally and is incorporated in our model. As the reviewer states, it is likely these stoichiometric requirements for oscillation are due to multimeric interactions.

Another major role for stoichiometry is on the period and amplitude of the oscillators. Stoichiometric variations within the range that support oscillation still lead to slight changes in period and amplitude (the spread we show in Figure 6c,d from our model compared to our experiments in Fig 4e,f). The stoichiometric relationship in the model is shown in section 7.5. Equation 2 and 3 and the experimental data that guided the relationship used in the model equations are shown in Supplementary Fig. S10.

We have modified the section where we discuss the stoichiometric relationship in the main text to emphasize the importance of stoichiometry and concentration in our experiments and our model:

The PTO-GUVs are categorized as either "oscillating" or "non-oscillating" based on established thresholds for minimum concentrations and stoichiometric ratios of the PTO in bulk solution. Oscillating PTO-GUVs for all loading concentrations and diameters are shown as green dots (**Fig. 6A**). The model shows that oscillating PTO-GUVs occupy the left half of the cloud with higher numbers of oscillating PTO-GUVs at the higher mean PTO concentration and almost no oscillating PTO-GUV at the lowest mean PTO concentration. A two-dimensional representation

of the model shows excellent correspondence of the fidelity of the simulated PTO-GUVs with the experimental PTO-GUVs (**Fig. 6B**). We also calculate the periods and amplitudes of each oscillating PTO-GUV using experimentally derived relationship between protein concentration and stoichiometry and plot the corresponding histograms in **Fig. 6C** and **Fig. 6D**. We provide details of the calculations in the Methods. The distribution of periods and amplitudes obtained from the model follows the distribution of experimental PTO-GUV periods and amplitudes well (compare **Fig. 6C, D** with **Fig. 4E, F**).

Reviewer #2 (Remarks to the Author):

Reviewer: The authors have adequately addressed my comments. I recommend publication

Reviewer #3 (Remarks to the Author):

Reviewer: *I thank the authors for their thoughtful consideration of my previous comments. I do think it would add something substantial to the paper to test experimentally the prediction that increasing [KaiB] only increase "fidelity" rather than just in a model. But if this is prohibitive due to personnel constraints (implied in the response letter), my judgment is that this is a valuable paper that is worth publishing already in its current form.*

Response: We thank the reviewer for their comments. We agree that performing the experiments could be useful but given the personnel constraints this is not feasible. We thank the reviewer for their kind comments and for supporting the publication of the paper in its current form.